# Extreme Li-Mg selectivity via precise ion size differentiation of polyamide membrane

Quan Peng[1,2,5], Ruoyu Wang [3,5], Zilin Zhao[2], Shihong Lin[3], Ying Liu[1,2], Dianyu Dong[2], Zheng Wang[2], Yiman He[2], Yuzhang Zhu [1,2] ✉, Jian Jin [1,2] ✉ & Lei Jiang [4]

Achieving high selectivity of Li[+] and Mg[2+] is of paramount importance for effective lithium extraction from brines, and nanofiltration (NF) membrane plays a critical role in this process. The key to achieving high selectivity lies in the on-demand design of NF membrane pores in accordance with the size difference between Li[+] and Mg[2+] ions, but this poses a huge challenge for traditional NF membranes and difficult to be realized. In this work, we report the fabrication of polyamide (PA) NF membranes with ultra-high Li[+]/Mg[2+] selectivity by modifying the interfacial polymerization (IP) process between piperazine (PIP) and trimesoyl chloride (TMC) with an oil-soluble surfactant that forms a monolayer at oil/water interface, referred to as OSARIP. The OSARIP benefits to regulate the membrane pores so that all of them are smaller than Mg[2+] ions. Under the solely size sieving effect, an exceptional Mg[2+] rejection rate of over 99.9% is achieved. This results in an exceptionally high Li[+]/Mg[2+] selectivity, which is one to two orders of magnitude higher than all the currently reported pressure-driven membranes, and even higher than the microporous framework materials, including COFs, MOFs, and POPs. The large enhancement of ion separation performance of NF membranes may innovate the current lithium extraction process and greatly improve the lithium extraction efficiency.

Salt lakes hold 60–80% minable lithium reserves of the world[1]. Exploring the efficient access to lithium resources in salt lakes is crucial for providing stable lithium supply to meet the rapidly growing demand of the lithium-ion battery industry. As for extracting lithium from brine, one of the key processes is to separate Li[+] and Mg[2+] ions since the other ions with higher abundance, such as Na[+], K[+], and Ca[2+], will be pre-removed in the form of precipitation through solar-driven evaporation[2–4]. Efficient Mg[2+]/Li[+] separation still faces significant challenges because the content of Mg[2+] in brine is much higher than that of Li[+] (for example, the content of Mg[2+] is usually more than 20 times of

Li[+] in salt lake brines in China[5]), and the sizes of Li[+] and Mg[2+] ions are very similar[6,7].

Pressure-driven nanofiltration (NF) technology based on membrane separation is one of the most concerned technologies among the commonly used treatment trains for extracting lithium from brine[8–11]. However, the current NF membranes exhibit relatively low Li[+]/Mg[2+] separation selectivity, usually less than 20. Therefore, in the process of lithium extraction, multiple repeated membrane separation processes have to be used to obtain sufficient purity of Li[+] in the filtrate to generate LiCO₃ or LiOH precipitation as the target product. The

[1]College of Chemistry, Chemical Engineering and Materials Science, Innovation Center for Chemical Science & Jiangsu Key Laboratory of Advanced Functional Polymer Design and Application, Soochow University, Suzhou 215123, PR China. [2]i-Lab, Suzhou Institute of Nano-Tech and Nano-Bionics, Chinese Academy of Sciences, Suzhou 215123, PR China. [3]Department of Civil and Environmental Engineering, Vanderbilt University, Nashville, TN 37235, USA. [4]CAS Key Laboratory of Bio-inspired Materials and Interfacial Science, Technical Institute of Physics and Chemistry, Chinese Academy of Sciences, 100190 Beijing, PR China. [5]These authors contributed equally: Quan Peng, Ruoyu Wang. ✉e-mail: yzzhu2011@suda.edu.cn; jjin@suda.edu.cn

limited selectivity of NF membranes is mainly caused by their uneven pore size and wide distribution[12–14]. Li$^+$/Mg$^{2+}$ separation is usually realized by co-effect of size sieving and Donnan exclusion[15–17]. Increasing the charge positivity of the membrane to enhance Donnan exclusion has been widely used to enhance the repulsion to Mg$^{2+}$ and thereby improve Li$^+$/Mg$^{2+}$ separation selectivity[18–20]. However, strong Donnan exclusion can also increase the repulsion to Li$^+$ ions, making it difficult to significantly increase the Li$^+$/Mg$^{2+}$ selectivity and simultaneously reducing the recovery rate of Li$^+$. Undoubtedly, if we can regulate the pores of the NF membrane to achieve Li$^+$/Mg$^{2+}$ separation solely relying on pore size sieving, it may break the limitation of charge repulsion, thus achieving a disruptive improvement of Li$^+$/Mg$^{2+}$ separation performance.

The preparation of the state-of-the-art NF membranes is based on interfacial polymerization (IP) reaction between amine monomer dissolved in water and acyl chloride monomer dissolved in organic solvent to form ultrathin polyamide (PA) selective layer[21–25]. During this process, the diffusion of monomers towards the interface (mainly amine monomers from aqueous phase to organic solvent) is slow and random[26–28], while the reaction rate between amine and acyl chloride is fast and the reaction is violent[29]. This leads to an uneven structure in the PA selective layer, resulting in a wide pore size distribution that is difficult to control[30]. To achieve precise regulation of the membrane pores and highly selective separation of target ions, an effective method is needed to control the IP process to improve the regularity of the reaction. Unfortunately, research in this area is still immature and there are few relevant reports. We previously reported a strategy of using water-soluble surfactants to form ordered monolayers at the water/oil interface to regulate the IP process (referred to as SARIP) to improve the pore size uniformity of PA NF membranes[31,32]. The resulting PA NF membranes with greatly narrowed pore size distribution exhibited a rejection rate of over 90% for all divalent cations. Although the SARIP strategy has shown its effective in regulating membrane pores, it is still difficult to meet the requirements for high selectivity separation of Li$^+$ and Mg$^{2+}$. The pore size range of the PA NF membrane prepared from SARIP is 3.0–8.8 Å (including all detectable pores). The hydration diameter of Mg$^{2+}$ ion is 8.6 Å, while the Stokes diameter of Li$^+$ ion is 4.8 Å (considering that Li$^+$ ion is prone to dehydration during pressure-driven separation due to its low hydration energy[33–35]). Obviously, Mg$^{2+}$ ions still can penetrate the membrane through partial pore, thereby creating insufficient Li$^+$/Mg$^{2+}$ selectivity (less than 50).

In the SARIP process, in addition to the surfactant monolayer at the water/oil interface, there are also a large number of free surfactant molecules and their aggregates such as micelles in the aqueous phase. The presence of these free surfactant molecules and aggregates will partially weaken the regulatory effect of surfactant monolayers on the interfacial diffusion of amine monomers, limiting the improvement of membrane pore uniformity[32]. In this work, we report the use of an oil-soluble surfactant (i.e. dodecyl phosphate (DDP)) introduced into the organic solvent phase to regulate IP reaction, referred to as OSARIP. The amphiphilic nature of DDP molecules enables them to form assembled monolayers at the water/oil interface. Due to the absence of free surfactant molecules in the aqueous solution, the DPP monolayer at the water/oil interface can fully exert its effect in facilitating amine monomer diffusion and regulating IP reaction. As expected, the OSARIP process makes the pores of the PA NF membranes more uniform, and the pore size distribution further narrowed to be in the range of 3.2–8.0 Å. This means that almost all pores are smaller than the hydration diameter of Mg$^{2+}$ ion, and most pores are larger than the Stokes diameter of Li$^+$ ion (considering that Li$^+$ ion is prone to dehydration during pressure-driven separation due to its low hydration energy[33–35]). Therefore, relying solely on the size sieving effect, the PA NF membrane exhibits ultra-high Mg$^{2+}$ rejection of up to 99.96% and ultra-high Li$^+$/Mg$^{2+}$ selectivity of up to 4147. The ultra-high separation

selectivity of Li$^+$/Mg$^{2+}$ makes it possible to obtain high-purity lithium filtrate through a single membrane filtration process. This may transform the existing technology flow of extracting lithium from brine, significantly simplifying the process and improving separation efficiency.

## Results

To regulate the IP reaction effectively, an oil-soluble surfactant, DDP, was introduced into the hexane solution. The amphiphilic nature of DDP allows for the formation of a monolayer at oil/hexane interface. The lowest interfacial tension was observed at a DDP concentration of 0.1 g L$^{-1}$ (see Supplementary Fig. 1), indicating the formation of the densest molecular layer. The presence of DDP monolayer can adsorb the PIP molecules to the water/hexane interface via electrostatic interaction between the phosphonic group in DDP and the amine group in PIP (Fig. 1a). This will cause an increase in PIP concentration near the oil/hexane interface, reaching its highest point at a DDP concentration of 0.1 g L$^{-1}$, as evidenced by UV-vis absorption spectra. (Fig. 1b, and Supplementary Fig. 2). Further in-situ monitoring of PIP absorbance near the oil/hexane interface during the IP reaction reveals a faster decrease of PIP absorbance when there is a DDP monolayer at interface (Fig. 1b). This indicates that the DDP monolayer is helpful in promoting the trans-interface diffusion of PIP monomers. In addition to the accumulation effect of DDP monolayer, free DDP molecules in hexane act as transport carriers through the dynamic exchange process with the monolayer molecules to assist in the trans-interface diffusion of PIP molecules[36]. Moreover, the DDP assembly significantly reduces the oil/water interface tension, which is beneficial for reducing the hindrance of PIP diffusion and enhancing the spatial uniformity of the diffusion process. It is important to note that the OSARIP process eliminates the presence of any free surfactant molecules or micelles in the aqueous solution, avoiding any competition with the surfactant assembly at interface and creating an ideal condition for unleashing the full potential of the surfactant assembly in controlling PIP trans-interface diffusion.

In the traditional IP process, the trans-interface diffusion of PIP is slow and random, making it difficult for the supply of PIP to keep up with its consumption. Therefore, the IP reaction exhibits spatial and temporal discontinuity, which ultimately leads to the chemical and structural heterogeneity of the PA membrane. In the OSARIP process, the diffusion of PIP is fast and spatially uniform, which greatly reduces the significant difference between monomer diffusion rate and IP reaction rate, thereby enhancing the continuity of PA layer formation in time and space. Compared to the PA selective layer prepared from traditional IP, the cross-linking degree of the PA selective layer prepared from OSARIP is significantly increased (from 74.7% to 88.6%) based on the elemental composition analysis of X-ray photoelectron spectroscopy (XPS) (Fig. 1c, d, and Supplementary Figs. 3–6), and the membrane shows less negatively charged due to less residual unreacted groups (see Supplementary Fig. 7).

Figure 1d–g displays the topological structure of the PA NF membrane prepared from both traditional IP and OSARIP. In comparison, the PA NF membrane prepared from OSARIP displays small convex spots and a thinner PA selective layer (~67 nm for the membrane prepared from traditional IP and ~55 nm for the membrane prepared from OSARIP). We further probed the pore size information of the membranes by conducting a standard analysis of the rejections of neutral molecules with different molecular weights. As the concentration of DDP in hexane increases, the molecular weight cut-off (MWCO) of the membrane gradually decreases and reaches a minimum value of 134 Da at a critical concentration of 0.1 g L$^{-1}$ (Fig. 2a). To our knowledge, such a small MWCO has not yet been achieved in the PA NF membrane prepared by the IP reaction of PIP and TMC reported so far. Further increasing the DDP concentration to 0.15 g L$^{-1}$ actually increases the MWCO of the membrane to 185 Da.

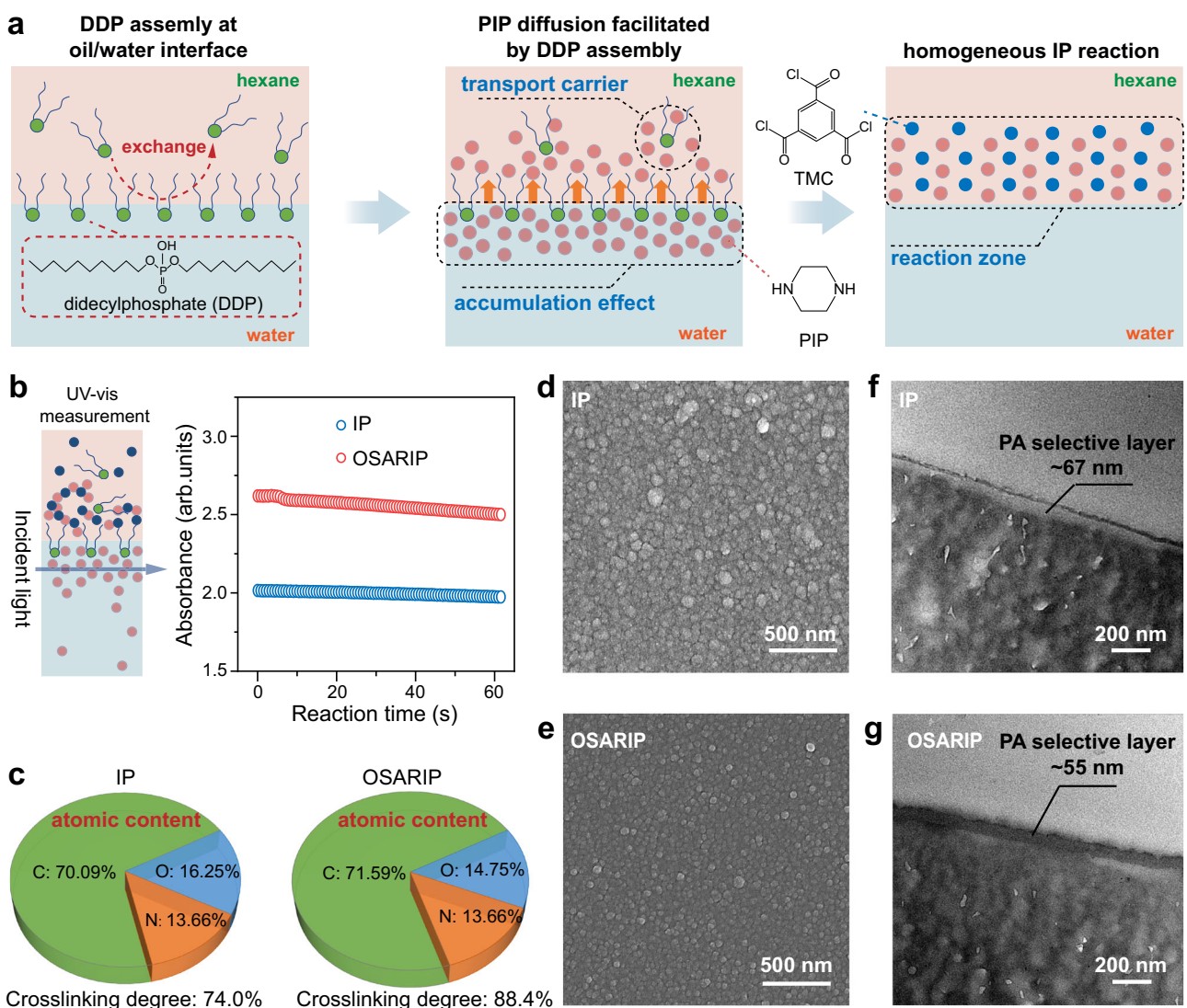

**Fig. 1 | Fabrication, chemical composition, and topological structure of PA NF membranes. a** Scheme for illustrating the effect of DDP assembly at the oil/water interface on PIP trans-interface diffusion to enhance the homogeneity of the IP reaction. **b** Variation of PIP absorbance near the interface with respect to the reaction time monitored by UV-vis measurement. **c** Chemical elemental composition and crosslinking degree of PA NF membranes prepared from traditional IP and OSARIP. **d–g** Surface and cross-sectional structure of PA membranes prepared from traditional IP and OSARIP (**d**, **e** Top-view SEM images. **f**, **g** Cross-sectional TEM images).

Following the log-normal distribution assumption, the pore size distribution of the membranes was fitted from the rejection curve of neutral molecules and expressed as a probability density function (PDF) (see Section 1.4.2 of Supplementary Information). In keeping with the trend of MWCO change, the introduction of DDP greatly narrows the pore size distribution of PA NF membranes and results in a narrowest pore size distribution with a shape parameter of 1.13 (defined as the geometric standard deviation of the PDF curve) at a DDP concentration of $0.1 \, g \, L^{-1}$ (Fig. 2b). Notably, the pore size falls in the range of the PA NF membrane prepared from the traditional IP, which indicates that the DDP monolayer primarily improves the uniformity of pore size, rather than reducing the pore size. These highly uniform pores enable the membrane with an ultra-high rejection towards all divalent salts including anionic salt of $Na_2SO_4$ and cationic salt of $MgCl_2$ (all over 99%) (Fig. 2c, Supplementary Fig. 8 and Supplementary Table 1), though the membrane is negatively charged. Interestingly, the pore structure and salt rejection of the PA NF membrane prepared from OSARIP at a DDP concentration of $0.1 \, g \, L^{-1}$ is little affected by the reaction time (see Supplementary Figs. 9 and 10),

which further proves the facilitating effect of DDP assembly on the IP kinetics.

Upon the analysis of existing NF membranes in the literatures (Supplementary Table 2), it is clear that NF membranes with small pore sizes are highly effective at rejecting $MgCl_2$ without relying much on Donnan exclusion (Fig. 2d). This discovery underscores the immense potential of employing such membranes for applications that demand efficient removal of divalent cations by size sieving (e.g., $Li^+/Mg^{2+}$ separation for Li extraction from brines). Compared with the existing NF membranes, our PA NF membrane possesses the smallest effective pore size and exceptionally narrow pore size distribution, thereby exhibiting the highest rejection to $MgCl_2$. It also exhibits very high rejection to other divalent cations, such as 98.36% for $Ba^{2+}$, 99.13% for $Mg^{2+}$, 99.03% for $Ca^{2+}$, and 99.21% for $Zn^{2+}$ (Supplementary Fig. 11 and Supplementary Table 3). Meanwhile, due to the dehydration process, the steric hinderance of the membrane pores cannot obstacle the penetration of monovalent cations such as $Li^+$, leading to a relatively low rejection (40.78%) to it. Consequently, a very sharp transition region is presented by the membrane when correlating the ion

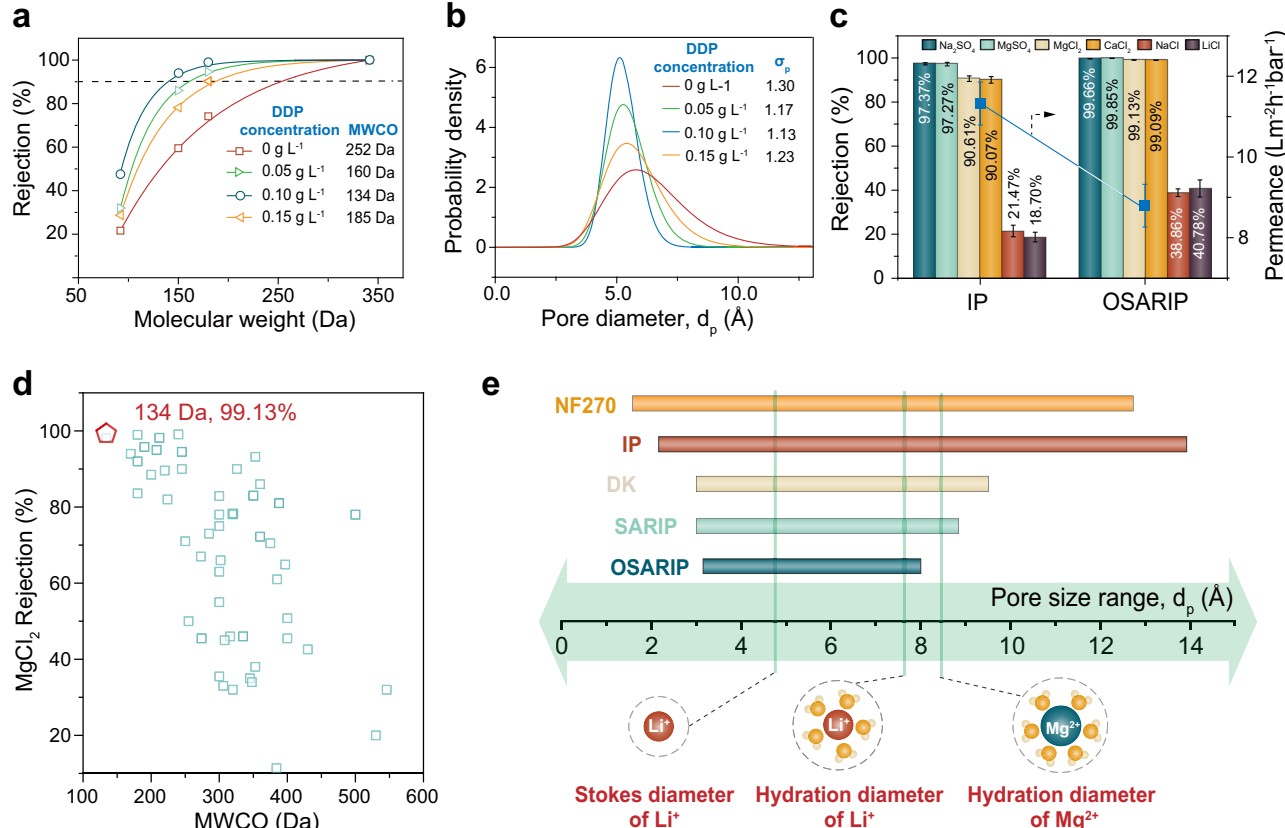

**Fig. 2 | Pore size and single salt rejection of PA NF membranes prepared from OSARIP. a** Rejection of neutral molecules by the membranes prepared with different concentration of DDP in hexane. **b** Pore size distribution of the corresponding membranes fitting from the rejection curve of neutral molecules following log-normal distribution principle. **c** Salt rejections of the membrane prepared from traditional IP and OSARIP. **d** Summary of $MgCl_2$ rejection by the PA membrane prepared from OSARIP at optimal condition (red pentagon) and the NF membranes reported by others (square, see detail description in Supplementary Table 2). **e** Comparison of pore size range of PA NF membranes prepared from traditional IP, SARIP, OSARIP, and the commercially available NF membranes (DK produced by Suez and NF270 produced by Dupont). All membranes are negatively charged based on the results of streaming potential measurement. Error bar represents the standard deviation of three replicate measurements.

rejections with their stokes radius (Supplementary Fig. 11), suggesting its superior ion-ion selectivity between the monovalent and divalent cations.

In order to emphasize the impressive capability of DDP assembly in refining pore size distribution, we compared the pore size ranges of several representative types of PA NF membranes including the NF membranes obtained from OSARIP, SARIP, and traditional IP processes, as well as commercially available DK and NF270 membranes (the pore size ranges derived from the PDF curves as reported by others[37]) (Fig. 2e and Supplementary Fig. 12). The pore size range of these membranes is directly taken from their PDF curves, where all pores with probability density greater than 0.01 are counted for statistics. It clearly shows that the PA NF membrane prepared from traditional IP process has a wide pore size range of 2.0–14.0 Å, followed by NF270 and DK membranes with pore size ranges of 1.6–12.6 Å and 3.0–9.5 Å, respectively. In contrast, the introduction of surfactants at the oil/water interface has shown a significant effect on sharpening the pore size range of membranes. Our previous work[31] demonstrated the efficacy of utilizing a water-soluble surfactant monolayer to regulate the IP reaction (e.g. SARIP), resulting in a PA NF membrane with a pore size range of 3.0–8.8 Å. Here, the newly developed OSARIP process further narrows the membrane pore size range to be 3.2–8.0 Å. Considering that the hydration diameter of $Mg^{2+}$ ion is 8.6 Å, it means that almost all the pores of the PA NF membrane prepared from OSARIP process are smaller than the hydration diameter of $Mg^{2+}$ ions, therefore resulting in an ultra-high rejection to $Mg^{2+}$ ions.

The $Li^+/Mg^{2+}$ separation performance of the PA NF membranes prepared from OSARIP process was further investigated by using a binary salt mixture composed of $MgCl_2$ and $LiCl$ with the total salt concentration of 2000 ppm and the $Li^+/Mg^{2+}$ mass ratio of 20:1 as simulated brine. It shows a rejection of $Mg^{2+}$ up to 99.82% (Fig.3a), while a negative rejection of −48.35% to $Li^+$. The corresponding $Li^+/Mg^{2+}$ selectivity is as high as 828. In comparison, the PA NF membrane prepared from traditional IP shows a rejection of 92.56% to $Mg^{2+}$ and −19.25% to $Li^+$, respectively, corresponding to a $Li^+/Mg^{2+}$ selectivity of 16.

The composition of ions in actual salt lake brine is diverse, so we further studied the separation performance of PA NF membrane under different $Mg^{2+}/Li^+$ ratios. When the total salt concentration remains unchanged at 2000 ppm and the $Mg^{2+}/Li^+$ mass ratio in the binary salt mixture increases from 10:1 to 60:1, the rejection to $Mg^{2+}$ slightly increases from 99.79% to 99.96% and the rejection to $Li^+$ decreases from −17.25% to −47.14% (Fig. 3a and Supplementary Table 4). This corresponds to the increase of $Mg^{2+}/Li^+$ selectivity from 548 to 4147. When maintaining a $Mg^{2+}/Li^+$ mass ratio at 20:1 and increasing the total salt concentration from 2000 to 5000 ppm, the rejection of $Mg^{2+}$ decreases slightly from 99.82% to 98.95%, while the rejection of $Li^+$ undergoes a significant change, decreasing from −48.35% to −94.35% correspondingly (Fig. 3b and Supplementary Table 5). This allows the PA NF membrane to still exhibit a high $Li^+/Mg^{2+}$ selectivity up to 185 in high salinity solution. In practical industrial applications of NF technology for lithium extraction from brine, feed solutions typically

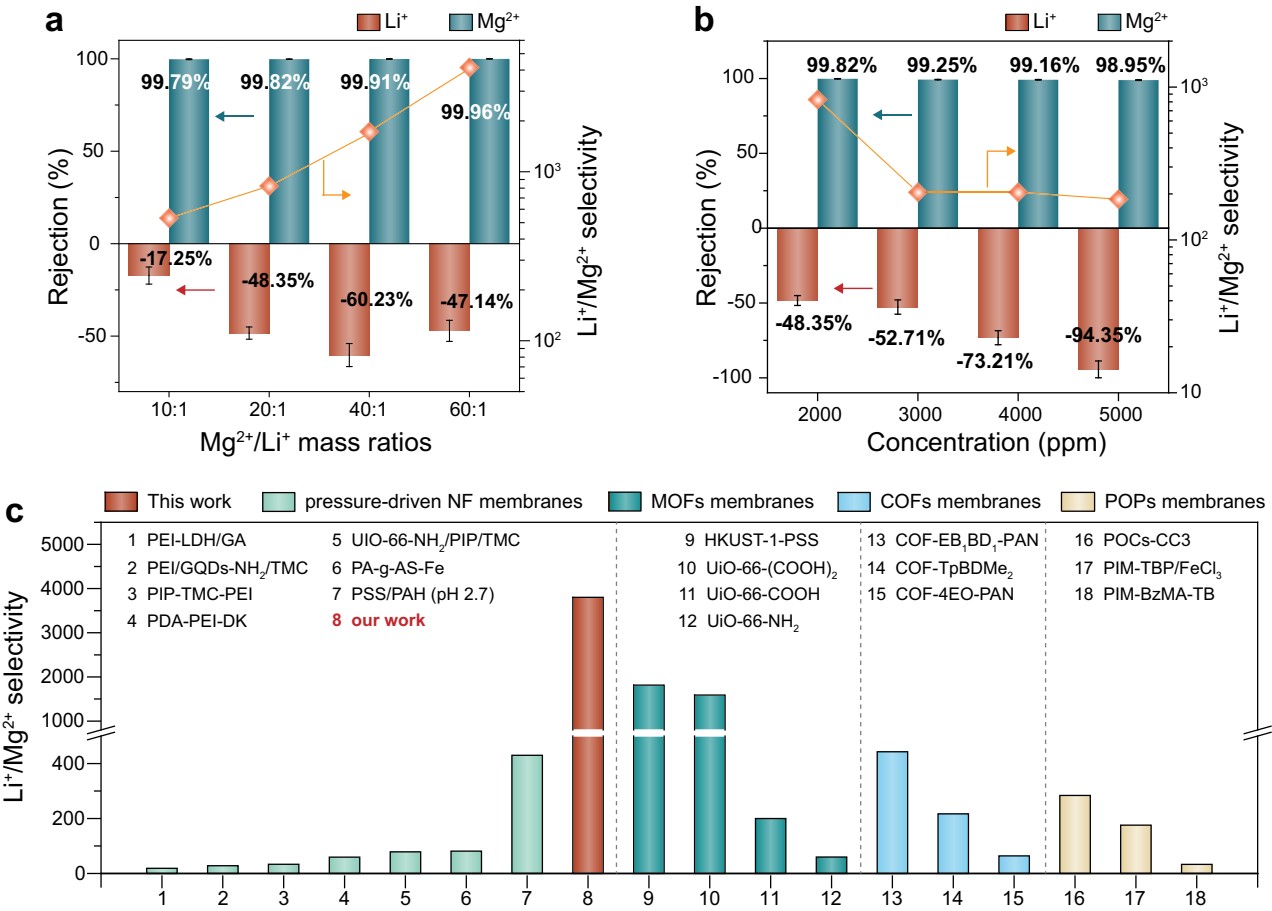

**Fig. 3 | Li⁺/Mg²⁺ separation performance of PA NF membranes by using a binary salt mixture of Li⁺ and Mg²⁺ as feed. a** Variation of the rejections of Mg²⁺ and Li⁺ and Mg²⁺/Li⁺ selectivity with respect to the Mg²⁺/Li⁺ mass ratio (The salt concentration of feed solution is 2000 ppm). **b** Variation of the rejections of Mg²⁺ and Li⁺ and Mg²⁺/Li⁺ selectivity with respect to the salt concentration of feed solution (The Mg²⁺/Li⁺ mass ratio is 20:1). **c** Comparison of Li⁺/Mg²⁺ selectivity of the PA NF membrane prepared in this work and the membranes reported by others including polymeric NF membrane, MOFs membranes, COFs membranes, and POPs membranes (see detail description in Supplementary Tables 7 and 8). Error bar represents the standard deviation of three replicate measurements.

exhibit higher concentrations and contain additional metal ions, such as Na⁺, K⁺, and Ca²⁺. Even under these conditions, the PA NF membrane demonstrates exceptional Li⁺/Mg²⁺ selectivity of more than 200 (Supplementary Figs. 13 and 14 and Supplementary Table 6). This remarkable performance underscores the unrivaled superiority of membranes with uniform pore sizes for lithium extraction from salt lake brine.

We compared the Li⁺/Mg²⁺ selectivity of PA NF membranes prepared from OSARIP with the most representative membranes reported so far, including pressure-driven NF membranes, microporous framework membranes such as covalent organic frameworks (COFs) membranes, metal-organic frameworks (MOFs) membranes, and porous organic polymers (POPs) membranes (Fig. 3c and Supplementary Tables 7 and 8). The Li⁺/Mg²⁺ selectivity of the PA NF membrane prepared from OSARIP is currently the highest value, which is one to two orders of magnitude higher than the pressure-driven NF membranes reported so far and even much higher than the emerging microporous framework membranes that are considered to have ideal uniform pore structures.

To further investigate the significant impact of pore structure on ion-ion selectivity of PA NF membranes, we prepared a series of membranes with the pore size range from wide to narrow by tailoring the DDP concentration in hexane within the scope of 0 to 0.125 g L⁻¹ (Fig. 4a, Supplementary Fig. 15 and Supplementary Tables 9 and 10). With narrowing the pore size range, the mean pore size decreases simultaneously. We studied the variation of Mg²⁺ and Li⁺ rejections and

Li⁺/Mg²⁺ selectivity over the mean pore size of as-prepared PA NF membranes using the binary salt mixture of MgCl₂ and LiCl with a total salt concentration of 2000 ppm and Mg²⁺/Li⁺ mass ratio of 20:1 as feed. The Mg²⁺ rejection gradually increases along with the decrease of the mean pore size and reaches 99.82% at last (Fig. 4b). However, the Li⁺ rejection exhibits an opposite trend. A more negative rejection is presented by the membrane with a smaller mean pore size. As the rejection of the two ions changes in opposite direction, the Li⁺/Mg²⁺ selectivity increases exponentially as the mean pore size decreases.

In order to account for the significant increase of Li⁺/Mg²⁺ selectivity by narrowing the pore size range, as well as the increasingly negative rejection of Li⁺, we utilized the solution-diffusion-electromigration (SDEM) model to describe the trans-membrane ion transport in a binary salt mixture. The SDEM model has been extensively used to explain the ion transport behaviors through reverse osmosis (RO)/NF/forward osmosis (FO) membranes[38,39]. By introducing the concept of virtual solution and ignoring the convective coupling between solvent and solute transport across the membrane, the ion transport is mainly driven by the gradients of concentration and electrostatic potential. The ion flux for species $i$, $J_i$, in the SDEM model can be described using the modified Nernst−Planck Eq. (1):

$$J_i = -P_i\left(\frac{dc_i}{dx} + z_i c_i \frac{d\varphi}{dx}\right) \qquad (1)$$

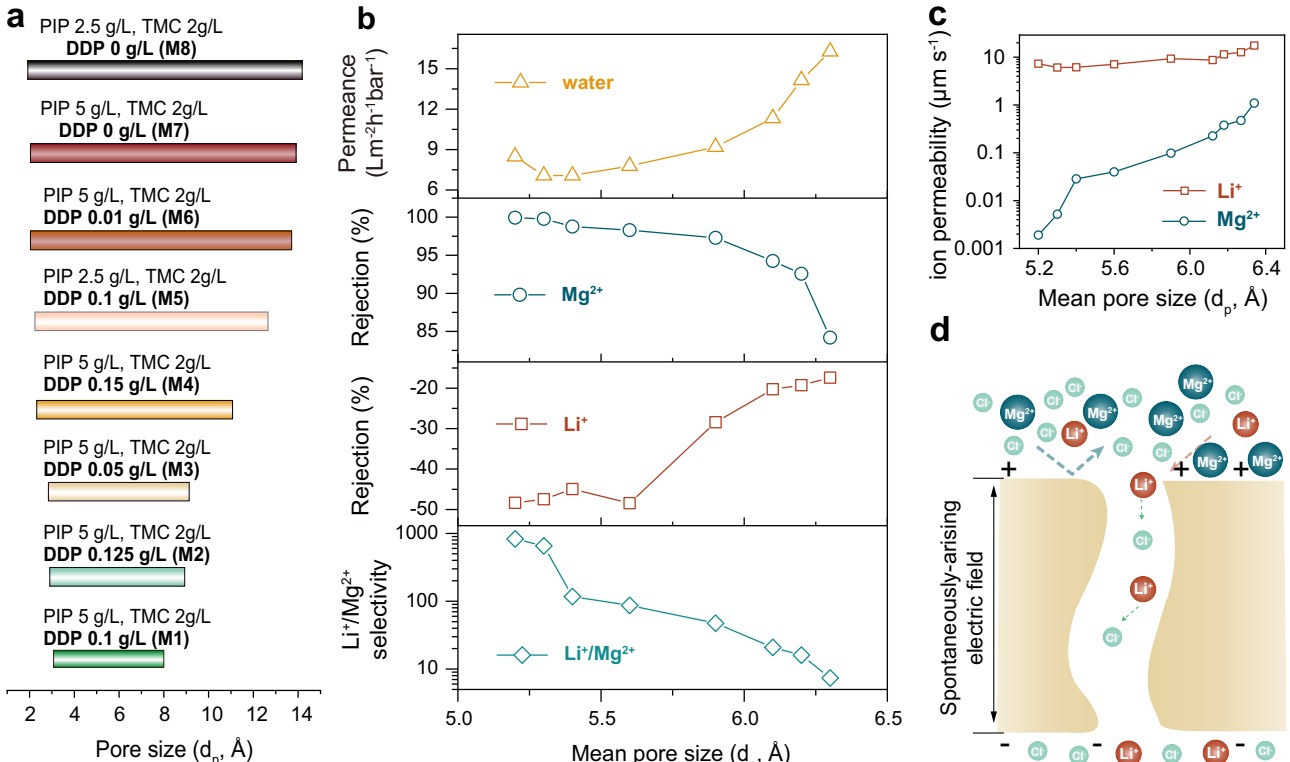

**Fig. 4 | Effect of pores of PA membranes on the separation performance for differentiating Li⁺ and Mg²⁺. a** Pore size range of PA NF membranes prepared with the conditions as listed in Supplementary Table 8 in detail. **b** Variation of water permeance, Mg²⁺ rejection, Li⁺ rejection, and Li⁺/Mg²⁺ selectivity over the mean pore size of as-prepared PA membranes. **c** Variation of ions permeability as function of mean pore size. **d** Schematic illustration of spontaneously-arising electric field during the separation process while using a binary salt mixture of MgCl₂ and LiCl as feed.

where $P_i$ is the ion permeability, $c_i$ is the ion concentration in a virtual solution that is in thermodynamic equilibrium with the membrane phase, $x$ is the trans-membrane coordinate normalized by the membrane thickness, $z_i$ is the valence of species $i$, and $\varphi$ is the dimensionless electrical potential in the virtual solution. Thus, the selective transport of Li⁺ and Mg²⁺ across the membrane is characterized by the difference of ion permeability in the SDEM model. Ion permeability, $P_i$, can be fitted from the experimental rejection results, given the feed composition and the measured permeate flux. As shown in Fig. 4c, the Li⁺ permeability can maintain at near 10 μm s⁻¹ with the decrease of the mean pore size due to the contribution of dehydration effect. In comparison, the Mg²⁺ permeability decreases over three orders of magnitude. The significant difference between ion permeability of Li⁺ and Mg²⁺ results in the ultra-high Li⁺/Mg²⁺ selectivity.

The change of Li⁺ rejection towards a more negative direction with the decrease of mean pore size can be attributed to the coupling effect between the trans-membrane ion transport of Li⁺ and Cl⁻ via a spontaneously-arising electric field, as presented schematically in Fig. 4d. Specifically, MgCl₂ is the dominant salt in the simulated brine and the concentration of Cl⁻ is mainly determined by the MgCl₂ concentration. Intercepting Mg²⁺ ions can lead to the accumulation of Cl⁻ ions near the membrane surface. As Mg²⁺ rejection increases, the concentration gradient of Cl⁻ across the membrane increases, resulting in the faster transport of Cl⁻. Accompanying the transport of Cl⁻, the transport of the counter-ion with equivalent charge is necessary to maintain charge neutrality in the permeate, thus establishing the spontaneous potential gradient. Considering the larger steric hindrance of Mg²⁺ for partition into the pores of the membrane, accelerating the Li⁺ transport is the only choice.

It is important to note that the transfer of ions through sub-nanometer pores is a highly intricate process influenced by various factors, including pore size, charge property, and chemical composition, as well as the coupling with water and other ions. In the case of binary salt solution comprising MgCl₂ and LiCl, the mass transfer behavior of Mg²⁺ and Li⁺ differs significantly. For the mass transfer of Mg²⁺, the highly uniform pores of the PA NF membrane can offer high and stable rejection of Mg²⁺ through a purely size-based separation. As for the mass transfer of Li⁺, the dehydration effect makes the pores of the PA NF membrane have limited effect to hinder the partition and transport of Li⁺ through the membrane, while the charge property and chemical structure of the membranes play an important role, as well as the effect of coupling with the counter-ion of Cl⁻. Negatively charged PA NF membranes are beneficial for obtaining high Li⁺ permeance due to the traction of Cl⁻. When the PA NF membranes are highly positively charged, the electrostatic attraction will slow down the migration of Cl⁻[40]. Meanwhile, the Donnan exclusion induced by the positive charge also produces strong hinderance for the partition of Li⁺ into the membrane pores. As a result, a higher rejection of Li⁺ is presented by the membrane, which is undesired for the target of recovering Li⁺. Firm evidence to prove this is that the PA NF membrane prepared from OSARIP by using polyethylene imine (PEI) and TMC as monomers, which are always considered to have strongly positive charge[41,42], displayed a much higher Li⁺ rejection as compared with the PA membrane prepared from OSARIP using PIP and TMC as monomers when using the same simulated brine as feed (Supplementary Fig. 16 and Supplementary Tables 11, 12). These results also indicate the superiority of negatively charged PA NF membranes with highly uniform pore size distribution for extracting Li⁺ from brine.

## Discussion
In summary, we developed a simple and effective method, referred to as OSARIP, to regulate the PIP trans-interface diffusion within the IP

reaction for the production of PA NF membranes with highly uniform pores. Through the utilization of the full promotion effect of oil-soluble surfactant monolayer at the oil/water interface on the PIP diffusion, the IP reaction between PIP and TMC results in a PA NF membrane with an ultra-narrow pore size distribution falling in the range between the hydration diameter of $Mg^{2+}$ ion and Stokes diameter of $Li^+$ ion. These highly uniform pores enable the membrane with an ultra-high rejection of >99% to $Mg^{2+}$ and relatively low rejection to $Li^+$. The performance test, using a binary salt mixture of LiCl and $MgCl_2$ to simulate the brine, revealed that the PA NF membrane prepared from OSARIP process display an ultrahigh $Li^+/Mg^{2+}$ selectivity over 4000. Further investigation into the ion transport behavior with the gradual decrease of effective pore size revealed that the steric hindrance has a limited effect on $Li^+$ permeability but greatly reduces the permeability of $Mg^{2+}$, resulting in the exponential increase of $Li^+/Mg^{2+}$ selectivity. Overall, our work offers a facile and precise method for optimizing the pore size of PA NF membranes at angstrom scale, which can be easily operated in the current industrial process for preparing NF membranes with highly improved performance for wide applications in water treatment and chemical separation.

## Methods

### Fabrication of PA NF membranes

Initially, aqueous solution containing 5 g/L PIP and hexane solutions containing 2 g/L TMC and a certain amount of dodecyl phosphate (DDP) were prepared, respectively. The two kinds of solutions were subsequently heated to 35 °C and kept at this temperature for further use. To fabricate the PA NF membranes via traditional IP, the top surface of the PES ultrafiltration membrane was first contacted with PIP aqueous solution for 60 s. Any visible aqueous drops on the membrane surface were then gently wiped away using a clean sheet. The TMC hexane solution containing DDP was then poured onto the surface of the PES and held horizontally for 30 s. The membrane was subsequently washed with pure hexane solution for 30 s to remove any unreacted TMC. The resulting membranes were cured in an oven at 60 °C for 30 min, and the membranes were finally washed with pure water and stored in water at 4 °C for further use. The process of creating PA NF membranes via traditional IP is similar to the fabrication of membranes via OSARIP but without the addition of DDP in hexane containing TMC.

### Membrane performance test

In this study, we utilized a cross-flow filtration system containing three circular cells to evaluate the separation performance of PA NF membranes. Before testing, the membrane was compacted using pure water at 6 bar for 1 h. Next, various salt solutions with a concentration of 1000 ppm were used as feed to pass through the membrane at 4 bar with a cross-flow rate of 50 L h$^{-1}$. The temperature of feed solution was maintained at 25 °C. To calculate the permeance of the PA NF membranes, we employed Eq. (2).

$$Permeance = \frac{\Delta V}{A \cdot \Delta t \cdot \Delta P} \quad (2)$$

where $\Delta V$ (L) is the permeate volume collected within the period of $\Delta t$ (h). $A$ (m$^2$) is the effective area of the membrane, which is $7.1 \times 10^{-4}$ m$^2$. $\Delta P$ (bar) is the applied pressure. The salt rejection, $R$ (%), of the membranes is calculated according to Eq. (3):

$$R = \left(1 - \frac{C_p}{C_f}\right) \times 100\% \quad (3)$$

where $C_p$ and $C_f$ is used to denote permeate and feed solution concentrations, respectively, which is determined by the conductivity measured by a conductivity meter (FE30K, Mettler Toledo).

For evaluating the performance of PA NF membrane for separating $Li^+$ from brine, binary salt mixtures of $MgCl_2$ and LiCl with various concentrations and various $Mg^{2+}/Li^+$ mass ratios were used as the feed solutions. The volume of the feed solution is 2000 ml. Within the experiment, the applied pressure is 4 bar, and the cross-flow rate is 50 L h$^{-1}$. The experiment is performed under a closed-loop recycling model, in which both retentate and permeate were continuously returned to the feed tank. Prior to the experiment, the filtration process was stabilized for over 30 minutes. Subsequently, 2 ml permeate were collected for the further characterization. The concentrations of $Mg^{2+}$ and $Li^+$ in the permeate and feed solutions were determined by inductively coupled plasma spectroscopy (ICP, Agilent 5100, Bruke). The protocol for measuring the concentration $Li^+$ and $Mg^{2+}$ in the solution is as follows:

Establishing calibration curves: Calibration curves for $Li^+$ and $Mg^{2+}$ are established using standard samples with concentrations ranging from 0.1 to 20 ppm for $Li^+$ and 0.1 to 10 ppm for $Mg^{2+}$.

Permeate analysis: Considering the significant difference in $Li^+$ and $Mg^{2+}$ concentrations in the solution, they are quantified separately. The permeate solution is generally analyzed directly without dilution. If the detected concentration falls within the calibration curve range, the data is accepted. Otherwise, the solution is diluted and re-measured.

Feed analysis: The feed solution is always diluted 10 times before analysis. The diluted solution is then measured following the same protocol as the permeate.

The rejections of $R_{Mg2+}$ and $R_{Li+}$ were determined by Eq. (3). The $Li^+/Mg^{2+}$ selectivity, $S_{Li, Mg}$, could be calculated through the rejection of $Li^+$ ($R_{Li+}$) and $Mg^{2+}$ ($R_{Mg2+}$), the permeation rates of $Li^+$ ($J_{Li}$) and $Mg^{2+}$ ($J_{Mg2+}$), or the concentrations of $Li^+$ and $Mg^{2+}$ ions in both the feed ($C_{f, Li}$, $C_{f, Mg}$) and permeate ($C_{p, Li}$, $C_{p, Mg}$) streams according to the Supplementary Eq. (S3):

$$S_{Li,Mg} = \frac{\frac{J_{Li}}{C_{f,Li}}}{\frac{J_{Mg}}{C_{f,Mg}}} = \frac{\frac{J_w \cdot C_{P,Li}}{C_{f,Li}}}{\frac{J_w \cdot C_{P,Mg}}{C_{f,Mg}}} = \frac{\frac{C_{P,Li}}{C_{f,Li}}}{\frac{C_{P,Mg}}{C_{f,Mg}}} = \frac{1 - \left(1 - \frac{C_{P,Li}}{C_{f,Li}}\right)}{1 - \left(1 - \frac{C_{P,Mg}}{C_{f,Mg}}\right)} = \frac{1 - R_{Li}}{1 - R_{Mg}} \quad (4)$$

where $J_w$ is the water flux.

## Data availability

All data generated in this study are provided in the article, Supplementary Information and Source Data file. All data are available from the corresponding author upon request. Source data are provided with this paper.

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

## Acknowledgements

This work was supported by the National Natural Science Foundation of China (21988102, J.J.), the National Key Research and Development Program of China (2022YFB3805900, J.J.), and the Key Research and Development Plan of Jiangsu Province (BE2022056, J.J.).

## Author contributions

Y.Z. and J.J. designed the experiments; Q.P. and Y.Z. performed the experiments including the membrane preparation, performance test and data organization; R.W. performed the theoretical analysis; Z.Z., Y.L., D.D, Z.W. and Y.H. performed the supplementary experiments; Y.Z., R.W., S.L. and J.J. contributed to writing the manuscript. L.J. reviewed the paper and provided revised suggestion. All authors discussed the results.

## Competing interests

The authors declare no competing interests.
