## [Peer Review File · Nature Communications]

Extreme Li-Mg Selectivity via Precise Ion Size Differentiation of Polyamide MembraneReviewers' Comments:

Reviewer #1:

Remarks to the Author:

This manuscript reports the fabrication of polyamide membranes with high Li/Mg selectivity. Considering the rapidly growing demand of lithium in the battery industry, the separation of Li/Mg from brine solution is an important research question. The team developed a novel process to regulate the interfacial polymerization reactions to enhance the membrane selectivity. By using an oil-soluble surfactant to tune the monomer diffusion at the oil-water interfaces, the resulting polyamide membranes showed narrow pore size distribution, which led to high size sieving selectivity. The authors performed an in-depth study of the interfacial polymerization reactions and demonstrated impressive separation selectivity. The results are certainly important breakthroughs and of broad interests, hence it would be suitable for publication in Nature Communications. This reviewer has some minor concerns about the characterization of membranes, methods and explanation of results. Hope these suggestions help improve the quality of the work.

1. The pore size distribution is a key parameter that determines the ion separation performance. In this study, the pore size distribution was mainly quantified by the filtration of charge-neutral molecules. Did the authors use other physical characterization techniques to quantify the pore size? The authors could try to make free-standing films and perform more in-depth physical analyses of the membranes.
2. As the authors proposed a new way for regulating the PIP diffusion and the reaction kinetics of the formation of polyamide membranes, it would be interesting to study the evolution of membrane structures as a function of time, for example, how the structure and properties of the polyamide membranes change as a function of time?
3. Given that the membrane showed record-high selectivity, the Mg ions in the feed and permeate concentrations are very different. For example, for Mg/Li ratio of 60:1, the concentration of Mg in the feed and permeate solutions was measured to be 535 and 0.19±0.07ppm, respectively. The lithium concentration increased slightly from 8.4 to 12.36 ppm. Clearly, the Mg concentrations in the feed and permeate were very different. Accurate quantification of the salt concentrations is important. In the supplementary methods, "The concentrations of Mg²⁺ and Li⁺ in the permeate and feed solutions were determined by inductively coupled plasma spectroscopy (ICP, Agilent 5100, Bruker). It would be helpful if the authors could provide detailed test methods, including the dilution protocols, calibration curves, and more raw data of ICP analysis, and the uncertainty analysis. Did the authors dilute the permeate solution to quantify both Li and Mg concentrations or did they dilute the solution and quantify Li and Mg separately? Could the authors provide more raw data of ICP measurements and the uncertainty analysis? Given that the membranes show record-high selectivity, it is important to ensure the accuracy of measurements.
4. I checked the methods in the main paper and supplementary information, but could not find the duration of the permeation tests. The authors should provide more experimental details, such as volume and concentration changes of the feed solution. Did the authors measure the dynamic change of the rejections and selectivity as a function of time? Given that the membranes showed low rejections of the lithium ions, the concentration of the feed solutions might have changed rapidly during the testing. The authors are recommended to provide more detailed measurements of the volume of feed solutions, salt concentrations of feed solutions, and their dynamic change during the tests.
5. It would also be good to provide a mass balance calculation for the cross-flow measurements, for example, the feed concentrations at the beginning of tests and end of tests, and the change of salt concentrations of the permeate solution.
6. It was very interesting to tailor the DDP concentration to tune the pore size distribution. Varying the DDP concentration could form membranes with tailored pore size and consequently evident enhancement of Li/Mg selectivity. This reviewer is curious about the DDP molecules, which are negatively charged and it was not clear whether the surfactant was trapped in the polyamide networks, which might influence the structure and properties of the membranes. The authors did not

explain this in the paper. Could the authors elaborate on this in detail?

Reviewer #2:

Remarks to the Author:

Please see the attached reviewer comments.

This work reported a novel oil-soluble surfactant monolayer promotion strategy to regulate the PIP trans-interface diffusion during the IP reaction for the fabrication of PA NF membranes with a narrow pore size distribution. The resultant membrane shows a high rejection of > 99% to Mg^{2+} and relatively low rejection to Li^+ , leading to a Li^+/Mg^{2+} selectivity over 4000 for a binary salt mixture of LiCl and $MgCl_2$. The Li-Mg selectivity is record high among the reported PA membranes. Some specific comments are listed and should be addressed before the acceptance of the manuscript.

(1) “The hydration diameter of Mg^{2+} ion is 8.6 Å, what is the Stokes diameter of Li^+ ion 4.8 Å (considering that Li^+ ion is prone to dehydration during pressure-driven separation due to its low hydration energy).” As for describing the size of ion, why the hydration diameter is used for Mg^{2+} , and the Stokes diameter is used for Li^+ ? Is there a correlation between the dehydration and the Stokes diameter? Also pay attention to the grammar of this sentence.

(2) The cross-linking degree of PA is obtained from the elemental composition analysis of XPS. Please give more details.

(3) How the pore size distribution of the membranes is fitted from the rejection curve of neutral molecules, and how is it related to the probability density function (PDF)? What is the definition of a shape parameter? How the mean pore size is calculated?

(4) For the Li^+/Mg^{2+} separation performance of the membrane, why the rejection of Li^+ is negative? How the ion rejection is corresponded to the ion selectivity?

(5) As shown in Fig.4c, with the increasing mean pore size, the Li^+ permeability increases slightly, while the Mg^{2+} permeability rises prominently. Since almost all pores are smaller than the hydration diameter of Mg^{2+} ion, and most pores are larger than the diameter of Li^+ ion, should the increasing ion permeability of Li^+ be more obvious than that of Mg^{2+} ?

(6) The coupling effect between the trans-membrane ion transport of Li^+ and Cl^- via a spontaneously-arising electric field is confusing. The PA membrane is negatively charged, which shows a stronger attraction towards Mg^{2+} than Li^+ , and a repulsion towards Cl^- . Because Mg^{2+} is a dominant ion in the feed, the increasing mean pore size from 5.2 to 6.4 Å should have a remarkable effect on the Mg^{2+} permeability.

Reviewer #3:

Remarks to the Author:

In this manuscript, Peng et al. presents the use of an oil-soluble surfactant, dodecyl phosphate (DDP), during interfacial polymerization (IP) to enhance the pore size uniformity, and therefore selectivity, of nanofiltration membranes for Li⁺/Mg²⁺ separation. Membranes for Li⁺/Mg²⁺ separation have gained interest due to their ability to extract lithium from geothermal brines but are currently limited by low selectivities. Previous works have found that the addition of surfactants in the aqueous phase promotes the diffusion of amine monomers across the aqueous/organic interface during IP in a process known as surfactant-assembly regulated IP (SARIP), resulting in polyamide layers with more uniform pore size distributions. In this study, Peng et al. added the surfactant DDP in the organic, rather than aqueous, phase in order to eliminate surfactant aggregation in the aqueous phase and further enhance amine diffusion in a modified organic-SARIP (OSARIP) process. They report that OSARIP resulted in a narrower pore size distribution and smaller mean pore size than IP and SARIP, resulting in over 99% rejections of divalent salts by size-sieving. Notably, in mixed-salt solutions of MgCl₂ and LiCl, OSARIP membranes demonstrated Li⁺/Mg²⁺ selectivities as high as 4147 due to the combined effects of high Mg²⁺ rejection from size-sieving and enhanced Li⁺ transport from counter-ion coupling with Cl⁻. Overall, this work presents a significant improvement in Li⁺/Mg²⁺ separation by implementing a simple modification to conventional IP. However, there are several details that are somewhat oversold and overinterpreted and should be addressed to make a good scientific contribution. I also believe that the authors need to be able to show what would happen under more realistic scenarios with these membranes.

Questions and comments

1. Absolute pore size distribution in terms of angstroms from a simple correlation of polymers that pass through a NF membrane is an oversimplification that I don't think should be highlighted so prominently in this paper. As the authors themselves mention "no steric and hydrodynamic interactions between these neutral solutes and the pores of the membranes", we know that this is not true at this scale of the pore and sugars going through the pores. Also normal hydrodynamics does not even apply. I know it would make the paper less easy to write but I really don't agree with all the size-based arguments here because of all this uncertainty. The authors talk about narrowing pore size distribution from 3.0 - 8.8 Å to 3.2 - 8.0 Å, I am not sure that we can be confident about this at all given the methods used. However, I do agree that the pore size is definitely narrowed as seen from the sharper MW cutoff from the sugar filtration experiments.
2. The 4000 selectivity is a little oversold. That value only occurs at very low lithium concentration and very low total salt concentrations (2000 ppm) which is totally impractical for any brine-based lithium separations.
3. Further there is a dramatic impact of salinity on lithium selectivity, I then really question the authors' assertion that the selectivity is solely size based. Their further assertion of chloride coupled high transport of lithium is also suspect in the light of a mixed salt with NaCl as described below.
4. I also wanted to comment on this paragraph from the perspective of what is seen and what is actually applicable in a realistic scenario.

Lines 270 -274 "Specifically, MgCl₂ is the dominant salt in the simulated brine and the concentration of Cl⁻ is mainly determined by the MgCl₂ concentration. Intercepting Mg²⁺ ions can lead to the accumulation of Cl⁻ ions near the membrane surface. As Mg²⁺ rejection increases, the concentration gradient of Cl⁻ across the membrane increases, resulting in the faster transport of Cl⁻. Accompanying the transport of Cl⁻, the transport of the counter-ion with equivalent charge is necessary to maintain charge neutrality in the permeate, thus establishing the spontaneous potential gradient. Considering the larger steric hindrance of Mg²⁺ for partition into the pores of the membrane, accelerating the Li⁺ transport is the only choice."

Except Lithium is definitely not the only choice, in an actual brine there will be much more sodium which has an even hydration energy and about the same size of lithium. Thus exaggerated lithium selectivity seen here will not occur in any realistic scenario. This should be acknowledged prominently. Further, there should be at least one set of experiments with actual brine type composition of the feed stream to see how this membrane would perform under more realistic scenarios where these anomalies with single and two salts mixed together are diluted away.

5. The high negative rejection for lithium is less believable for me. Did the authors account for concentration polarization at all. It seems to me that the negative rejection goes up as the concentration of the Lithium goes down. Essentially there is less concentration polarization and thus less passage. Assuming a concentration polarization of 1.5-2x (which is not unreasonable based on crossflow rates), from table S4, none of the rejection values would be negative if the membrane concentration is used instead of the feed concentrations.
6. To confirm reproducibility, I would like to ask authors to include information on the number of membrane replicates made, number of trials, and error bars both in the main text and the supplementary information.
7. The authors reference the supplementary information for additional details and sources in the figures that could be included in the main text as well. Specifically, I would like to ask the authors cite the references for the data points shown in Figures 2-3. Additionally, Figure 4a would benefit from added details differentiating the eight prepared membranes, especially as M4 and M8 both are described as "DDP 0.1 g/L" yet exhibit significantly different pore size ranges.
8. On page 5, line 145, authors mention that the OSARIP membrane displayed a smoother surface compared to the IP one, referencing the surface and cross-sectional images shown in Figure 1. I would like to ask the authors if they have any further characterization of membrane roughness to justify this claim.
9. On pages 10-11, lines 289-300, the authors compare negatively-charged OSARIP membranes made by piperazine (PIP) to positively-charged ones made by polyethylene imine (PEI) to justify the claim that a negatively-charged surface enhances the Li⁺/Mg²⁺ selectivity of OSARIP membranes. However, membranes made by PEI tend to have looser structures compared to those made by PIP, and this was evident in the pore size distribution curves and performance data presented in the SI that show that PEI OSARIP membranes had slightly higher pore sizes, wider pore size distributions, and lower Mg²⁺ rejections than those made by PIP [1]. The authors suggested that the higher rejection of Li⁺ by PEI membranes was caused by increased Donnan exclusion from the positively-charged surface, but it seems possible that the higher Li⁺ rejection could've also been caused by a decreased coupling effect from lower Mg²⁺ rejections. It is thus not clear whether the difference in performance between PIP and PEI-based membranes should be attributed to their surface charges or difference in structures. Given this, would the authors be able to provide further experimental or simulation evidence to validate the claim that negatively-charged OSARIP membranes demonstrate higher selectivities?
10. One of the main findings in this work was that a concentration of 0.1 g/L DDP corresponded to the lowest interfacial tension as well as smallest mean pore size and distribution relative to both lower and higher concentrations. Interestingly, membranes with 0.1 g/L DDP also demonstrated both higher rejections and permeances relative to other concentrations, as shown in Figure S5, but this is not discussed in detail in the manuscript. I would like the authors to comment on the effect of surfactant concentration on OSARIP membranes.

[1] Wu, D., Yu, S., Lawless, D., & Feng, X. (2015). Thin film composite nanofiltration membranes fabricated from polymeric amine polyethylenimine imbedded with monomeric amine piperazine for enhanced salt separations. *Reactive and Functional Polymers*, 86, 168-183.

Response to Reviewer 1

General comment: This manuscript reports the fabrication of polyamide membranes with high Li/Mg selectivity. Considering the rapidly growing demand of lithium in the battery industry, the separation of Li/Mg from brine solution is an important research question. The team developed a novel process to regulate the interfacial polymerization reactions to enhance the membrane selectivity. By using an oil-soluble surfactant to tune the monomer diffusion at the oil-water interfaces, the resulting polyamide membranes showed narrow pore size distribution, which led to high size sieving selectivity. The authors performed an in-depth study of the interfacial polymerization reactions and demonstrated impressive separation selectivity. The results are certainly important breakthroughs and of broad interests, hence it would be suitable for publication in Nature Communications.

Response: We appreciate the positive comment given by the reviewer.

Comment 1: The pore size distribution is a key parameter that determines the ion separation performance. In this study, the pore size distribution was mainly quantified by the filtration of charge-neutral molecules. Did the authors use other physical characterization techniques to quantify the pore size? The authors could try to make free-standing films and perform more in-depth physical analyses of the membranes.

Response: Directly determining the pore size distribution of nanofiltration (NF) membranes remains a challenge. Transmission electron microscopy (TEM) and atomic force microscopy (AFM) struggle to resolve the sub-nanometer pores characteristic of these membranes, further complicated by the need for measurements in a water-saturated state. While positron annihilation lifetime spectroscopy (PALS) offers insights into free volume and size distribution within the polyamide active layer, its accessibility is limited. Therefore, a widely adopted alternative involves utilizing a series of charge-neutral molecules with varying molecular weights as probes. The reliability of this method is evidenced by its agreement with PALS measurements (see *Nat. Commun.* 2020, 11, 2015). Notably, while all pore size distributions in this study are derived from charge-neutral molecule rejection, the comparison between NF membranes is based on a consistent benchmark, ensuring the validity of the conclusions drawn.

While some have proposed analyzing the membrane further using free-standing

films, this approach may not accurately reflect the actual properties of the NF membrane. The significant influence of the support on its formation and transport processes cannot be ignored. Direct peeling of the active layer or dissolving the support with organic solvents to create a free-standing film inevitably damages its integrity. Similarly, in-situ polymerization at the oil/water interface without the support yields films that differ from those formed on the support due to the crucial role the support plays in monomer diffusion and interface state during interfacial polymerization.

Therefore, we advocate for directly characterizing the PA NF membrane itself as the most accurate approach. To gain a deeper understanding of the PA membrane prepared from OSARIP, we employed XPS measurements combined with Ar ion sputtering to elucidate its in-depth chemical elemental composition. As depicted in Figure R1, the nitrogen (N) atomic content exhibits a gradual increase with increasing etching time, while the oxygen (O) content shows a corresponding decrease. This trend suggests that the inner space of the PA active layer possesses a higher concentration of amine groups compared to the surface. Notably, the O/N atomic ratio within the inner space is less than 1, significantly lower than the surface, indicating a predominance of amine-terminated chemical structures. Prolonging the etching time to 360 seconds leads to the detection of a small amount of sulfur (S), attributed to the underlying PES support. At this depth, the N content slightly diminishes, while the O content rebounds. These collective observations point towards a heterogeneous structure within the PA layer along the depth axis. Based on the change in the O/N atomic ratio with respect to etching time, the inner part of the membrane appears to be denser than the front and back surfaces. The result of in-depth analysis of the PA membrane was added to the revised supplemental information as Supplemental Figure 7.

Figure R1. In-depth analysis of C, N, and O content in the PA active layer using the combination of XPS measurement and Ar ion sputtering technique.

Comment 2: As the authors proposed a new way for regulating the PIP diffusion and the reaction kinetics of the formation of polyamide membranes, it would be interesting to study the evolution of membrane structures as a function of time, for example, how the structure and properties of the polyamide membranes change as a function of time?

Response: In response to the reviewer's suggestion, we conducted a supplemental experiment to investigate the impact of reaction time on the structure and properties of the polyamide (PA) membrane prepared via OSARIP. The reaction time was changed from 10 s to 30 s. To assess the relationship between reaction time and membrane characteristics, we analyzed the pore size distribution of the PA membrane using the rejection of charge-neutral molecules, and the desalination performance of the resulting PA NF membranes. Figure R2 and R3 present the obtained results.

The surfactant assembly can significantly accelerate the IP reaction, enabling the PA membrane prepared from OSARIP with a 10 s reaction time to exhibit a remarkably low molecular weight cut-off (MWCO) of 154 Da (Figure R2a). This value is considerably smaller compared to the PA membrane prepared from traditional IP with a 30 s reaction time. Furthermore, the pore size distribution of the surfactant-assisted membrane was notably narrower (Figure R2b). Increasing the reaction time to 20 and 30 s yielded PA NF membranes with MWCO of 145 Da and 134 Da, respectively, suggesting that prolonged the reaction did not significantly affect the pore structure of the membrane prepared from OSARIP (Figure R2c-2f).

Consistent with the minimal pore size variation, the three OSARIP-derived PA

NF membranes exhibited comparable salt rejection performance (Figure R3a). Notably, they achieved very high rejection (around 99%) for all divalent salts tested, including Na_2SO_4 , MgSO_4 , and MgCl_2 . In contrast, the PA membrane prepared from traditional IP displayed a gradual increase in salt rejection with increasing reaction time. For instance, MgCl_2 rejection rose from 60.4% to 91.3% as the reaction time extended from 10 to 30 s (Figure R3b). This minimal deviation in pore size and salt rejection across different reaction times for the membranes prepared from OSARIP indicates a rapid transition to a self-limited polymerization state. This observation further proves the promoting effect of oil-soluble surfactant assembly at the oil/water interface on the IP kinetics. These results were added to the revised supplementary information as Supplementary Figure 9 and 10.

Figure R2. MWCO and pore size distribution of PA membranes prepared from

OSARIP with the reaction time of (a and b) 10 s, (c and d) 20 s, and (e and f) 30 s.

Figure R3. Salt rejection of PA membranes prepared from (a) OSARIP and (b) conventional IP with different reaction time.

Comment 3: Given that the membrane showed record-high selectivity, the Mg ions in the feed and permeate concentrations are very different. For example, for Mg/Li ratio of 60:1, the concentration of Mg in the feed and permeate solutions was measured to be 535 and 0.19 ± 0.07 ppm, respectively. The lithium concentration increased slightly from 8.4 to 12.36 ppm. Clearly, the Mg concentrations in the feed and permeate were very different. Accurate quantification of the salt concentrations is important. In the supplementary methods, “The concentrations of Mg^{2+} and Li^+ in the permeate and feed solutions were determined by inductively coupled plasma spectroscopy (ICP, Agilent 5100, Bruker). It would be helpful if the authors could provide detailed test methods, including the dilution protocols, calibration curves, and more raw data of ICP analysis, and the uncertainty analysis. Did the authors dilute the permeate solution to quantify both Li and Mg concentrations or did they dilute the solution and quantify Li and Mg separately? Could the authors provide more raw data of ICP measurements and the uncertainty analysis? Given that the membranes show record-high selectivity, it is important to ensure the accuracy of measurements.

Response: Following the reviewer's suggestion, we have added the ICP measurement protocol to the Methods section of the revised Supplementary Information. Briefly, the protocol involves:

Calibration curves: Calibration curves for Li^+ and Mg^{2+} are established using standard samples with concentrations ranging from 0.1 to 20 ppm for Li^+ and 0.1 to 10 ppm for Mg^{2+} .

Permeate analysis: Considering the significant difference in Li^+ and Mg^{2+} concentrations in the solution, they are quantified separately. The permeate solution is generally analyzed directly without dilution. If the detected concentration falls within the calibration curve range, the data is accepted. Otherwise, the solution is diluted and re-measured.

Feed analysis: The feed solution is always diluted 10 times before analysis. The diluted solution is then measured following the same protocol as the permeate.

Based on this established protocol, we are confident that the measured concentrations of Li^+ and Mg^{2+} are reliable.

Comment 4: I checked the methods in the main paper and supplementary information, but could not find the duration of the permeation tests. The authors should provide more experimental details, such as volume and concentration changes of the feed solution. Did the authors measure the dynamic change of the rejections and selectivity as a function of time? Given that the membranes showed low rejections of the lithium ions, the concentration of the feed solutions might have changed rapidly during the testing. The authors are recommended to provide more detailed measurements of the volume of feed solutions, salt concentrations of feed solutions, and their dynamic change during the tests.

Response: In this work, the separation performance of Li^+ and Mg^{2+} was evaluated under a closed-loop recycling model, in which both retentate and permeate were continuously returned to the feed tank. Prior to the experiment, the filtration process was stabilized for over 30 minutes. Subsequently, 2 ml permeate were collected for further characterization. Considering the significantly larger volume of the feed solution (2000 ml), the impact of collecting these small aliquots on the overall feed concentration is deemed negligible. This detail of performance measurement has been added to the Supplementary methods section in the revised Supplementary Information.

The reviewer proposed that the potential for rapid change of feed solution concentration due to the strong negative rejection of Li^+ and high rejection of Mg^{2+} . To verify this, we implemented the reviewer's suggestion to monitor changes in feed and permeate concentrations, as well as Li^+ and Mg^{2+} rejections, with respect to permeate volume. These results are presented in Figures R4 and R5. Figure R4a clearly demonstrates a decrease in feed Li^+ concentration with increasing water

recovery ratio (defined as the volume of collected permeate divided by the initial volume of the feed solution), which is induced by the negative rejection of Li^+ . Conversely, the high Mg^{2+} rejection leads to a rapid increase in its feed concentration, consequently raising the $\text{Mg}^{2+}/\text{Li}^+$ mass ratio. As previously shown in Figure 3a in the manuscript, higher $\text{Mg}^{2+}/\text{Li}^+$ mass ratios lead to stronger negative Li^+ rejection. This explains the observed increase in Li^+ concentration in the permeate with increasing water recovery ratio (Figure R4b). Additionally, the rising Mg^{2+} feed concentration is prone to increase its permeation, leading to a higher Mg^{2+} concentration in the permeate as well. Due to the dynamic changes in Li^+ and Mg^{2+} concentration in both feed and permeate, the cumulative rejections of Li^+ changed from -102.8% to -137.1% with an increase in water recovery ratio from 0 to 15% (Figure R5). At the same time, the cumulative rejection of Mg^{2+} changed from 99.3% to 98.8%. This suggests that a cumulative $\text{Li}^+/\text{Mg}^{2+}$ selectivity of 197 is obtained with a water recovery ratio of 15%. This selectivity is still higher than that achieved by the vast majority of other PA NF membranes, indicating the exceptional superiority of the PA NF membrane prepared from OSARIP for the separation of Li^+ and Mg^{2+} . We have added the Figure R4 and Figure R5 to the revised supplementary information as Supplementary Figure 13.

Figure R4. The concentration of Mg^{2+} and Li^+ in the (a) feed and (b) permeate as a function of water recovery ratio. This ratio is defined as the volume of collected permeate divided by the initial volume of the feed solution. The total salt concentration of the feed solution is about 10000 ppm and the $\text{Mg}^{2+}/\text{Li}^+$ mass ratio of this feed solution is 20:1. The operation pressure is 6 bar.

Figure R5. Mg²⁺ rejection and Li⁺ rejection as a function of water recovery ratio.

Comment 5: It would also be good to provide a mass balance calculation for the cross-flow measurements, for example, the feed concentrations at the beginning of tests and end of tests, and the change of salt concentrations of the permeate solution.

Response: In response to the reviewer's suggestion, we investigated the change in Li⁺ and Mg²⁺ concentrations in both the permeate and feed solutions at the beginning and end of the tests. Figure R4 presents the corresponding data. Further analyzing these changes, we performed a mass balance calculation, neglecting the minor influence of aliquots. The results are displayed in Figure R6. The total mass of Li⁺ and Mg²⁺ in the permeate and retentate combined was essentially identical to the mass in the feed solution, with a minor discrepancy of around 1%. We attribute this minor difference to the aforementioned aliquots. Figure R6 was added to the revised supplementary information as Supplementary Figure 14.

Figure R6. The mass of Li⁺ and Mg²⁺ in the feed, permeate and retentate.

Comment 6: It was very interesting to tailor the DDP concentration to tune the pore size distribution. Varying the DDP concentration could form membranes with tailored

pore size and consequently evident enhancement of Li/Mg selectivity. This reviewer is curious about the DDP molecules, which are negatively charged and it was not clear whether the surfactant was trapped in the polyamide networks, which might influence the structure and properties of the membranes. The authors did not explain this in the paper. Could the authors elaborate on this in detail?

Response: In response to the reviewer's suggestion, XPS measurement coupled with Ar ion sputtering was employed to investigate the depth profile of elemental composition, as shown in Figure R7. Notably, even as the etch time increased until the sulfur (S) signal associated with the PES support became evident, no phosphorus (P) signal was detected. This indicates that no surfactant was trapped within the polyamide (PA) network.

Figure R7. XPS survey spectra of PA membrane prepared from OSARIP with different etching time.

Response to Reviewer 2

General Comment: This work reported a novel oil-soluble surfactant monolayer promotion strategy to regulate the PIP trans-interface diffusion during the IP reaction for the fabrication of PA NF membranes with a narrow pore size distribution. The resultant membrane shows a high rejection of > 99% to Mg^{2+} and relatively low rejection to Li^+ , leading to a Li^+/Mg^{2+} selectivity over 4000 for a binary salt mixture of LiCl and $MgCl_2$. The Li-Mg selectivity is record high among the reported PA membranes.

Response: We appreciate the positive comment given by the reviewer.

Comment 1: “The hydration diameter of Mg^{2+} ion is 8.6 Å, what is the Stokes diameter of Li^+ ion 4.8 Å (considering that Li^+ ion is prone to dehydration during pressure-driven separation due to its low hydration energy).” As for describing the size of ion, why the hydration diameter is used for Mg^{2+} , and the Stokes diameter is used for Li^+ ? Is there a correlation between the dehydration and the Stokes diameter? Also pay attention to the grammar of this sentence.

Response: Firstly, we apologize for the grammatical errors. The sentence has been corrected to “The hydration diameter of Mg^{2+} ion is 8.6 Å, while the Stokes diameter of Li^+ ion is 4.8 Å (considering that Li^+ ion is prone to dehydration during pressure-driven separation due to its low hydration energy)” in the revised manuscript (page 3, line 27).

Regarding the reviewer’s concerning that why the hydration diameter is used for Mg^{2+} and the Stokes diameter is used for Li^+ , we would like to give a detail explanation as follows: We all know that the true size of ions in solution plays a crucial role in their transport through membranes. Ions generally exist in a hydrated state in aqueous solutions, and the hydration energy of different ions varies. In the case of Mg^{2+} , its hydration energy is high, which creating a stable hydration layer that is hard to destroy during its transport through membranes. Therefore, its hydration diameter is used as its effective size during the separation process. In contrast, Li^+ possesses a low hydration energy. When it is about to enter the membrane pore, its hydration layer is easily removed under pressure. Therefore, the Stokes diameter is usually used as the effective size for Li^+ (*Desalination* 2007, 214, 150-166; *J. Phys. Chem. B* 2010, 114, 3510; *J. Membr. Sci.* 2022, 648, 120358; *Nat. Commun.* 2020, 11, 2015).

Comment 2: The cross-linking degree of PA is obtained from the elemental composition analysis of XPS. Please give more details.

Response: According to the reviewer’s suggestion, we have added the XPS survey spectra of PA membrane obtained from traditional IP and OSARIP in the revised supplementary information as Supplementary Figure 3. The detailed surface atom composition and corresponding cross-linking degree of the membranes have been signed in the graph. The cross-linking degree was calculated according to the O/N atomic ratio of the membrane followed method reported previously (*Desalination* 2009, 242, 149–167; *Science* 2015, 348, 1347–1351). Corresponding illustration for

calculating the cross-linking degree has been added to the Methods section of Supplementary Information.

Supplementary Figure 3. Surface atomic composition of the PA membranes prepared from conventional IP and OSARIP determined by the XPS survey spectra.

Comment 3: How the pore size distribution of the membranes is fitted from the rejection curve of neutral molecules, and how is it related to the probability density function (PDF)? What is the definition of a shape parameter? How the mean pore size is calculated?

Response: The pore size distribution of the membranes is fitted from the rejection curve of neutral molecules. The detailed method is as follow: The neutral molecules used here included glycerol, xylose, glucose, sucrose and raffinose. The rejection of these neutral molecules by the as-prepared PA membranes was measured at 4 bar. With the premise of no steric and hydrodynamic interactions between these neutral solutes and the pores of the membranes, the corresponding pore size distribution can be expressed by the probability density function (PDF) (Eq.1).

$$\frac{DR(d_p)}{dd_p} = \frac{1}{r_p \ln \sigma_p \sqrt{2\pi}} \exp \left[-\frac{(\ln d_p - \ln \mu_p)^2}{2(\ln \sigma_p)^2} \right] \quad (1)$$

where d_p is the Stokes diameter of the neutral molecules that can be calculated according to Eq.2.

$$\text{Lg} \frac{d_p}{2} = -1.4962 + 0.4654 \text{lg} M \quad (2)$$

μ_p is the mean pore size, which is equals to the d_p of the neutral solute with a rejection of 50%. σ_p is the geometric standard deviation of the PDF curve and represents the distribution of the membrane pore size, which is the ratio of d_p of the

neutral molecule with a rejection of 84.13% to that of 50%. The σ_p is defined as the shape parameter.

The method for determining the pore size information of PA membranes by analyzing the rejection of neutral molecules has been explained in Methods section 1.4.2 of the Supplementary Information. We have also referenced this section in the revised manuscript on page 6, line 15.

Comment 4: For the $\text{Li}^+/\text{Mg}^{2+}$ separation performance of the membrane, why the rejection of Li^+ is negative? How the ion rejection is corresponded to the ion selectivity?

Response: For the negative rejection of Li^+ , it is attributed to a complex coupling effect between cation and anion transport through the membrane. Specifically, in binary salt solutions of MgCl_2 and LiCl , three ions (Mg^{2+} , Li^+ , and Cl^-) are present. During separation, each ion travels individually but is co-dependent to maintain electrical neutrality. In the binary salt solution, the Cl^- concentration is typically much higher than Li^+ . Since the PA membrane has limited ability to block monovalent ions due to their size and charge, Cl^- readily enters the pores and permeates through. This Cl^- transport generates a spontaneous electric field, which enhances the transport of Li^+ across the membrane to maintain electrical balance on the permeate side. This coupled effect significantly reduces Li^+ rejection, even leading to negative values, a phenomenon well-documented in NF processes involving multi-component feeds. (see *J. Membr. Sci.* 2001, 185, 223; *Chem. Eng. Sci.* 2013, 104, 1107; *Desalination* 2019, 468, 114081; *J. Membr. Sci.* 2021, 620, 118862; *Nat. Water* 2023, 1, 291).

For the question “How the ion rejection is corresponded to the ion selectivity”, the ion selectivity of Li/Mg could be calculated as follow:

$$S = \frac{1 - R_{\text{Li}}}{1 - R_{\text{Mg}}} = \frac{C_{p,\text{Li}}/C_{f,\text{Li}}}{C_{p,\text{Mg}}/C_{f,\text{Mg}}}$$

Where R_{Li} is the rejection of Li^+ , R_{Mg} is the rejection of Mg^{2+} . We have also added this calculation method to Eq. S3 in the revised Supplementary Information.

Comment 5: As shown in Fig.4c, with the increasing mean pore size, the Li^+ permeability increases slightly, while the Mg^{2+} permeability rises prominently. Since almost all pores are smaller than the hydration diameter of Mg^{2+} ion, and most pores are larger than the diameter of Li^+ ion, should the increasing ion permeability of Li^+

be more obvious than that of Mg^{2+} ?

Response: The hydration energy of an ion significantly influences its transport across membranes. Li^+ exhibits low hydration energy and readily sheds its hydration shell during separation process. Consequently, during transmembrane transport, Li^+ can adjust its hydration state according to the pore size. This flexibility minimizes pore size restrictions on its diffusion, resulting in only a moderate increase in Li^+ permeability with increasing pore size. In contrast, Mg^{2+} possesses a much higher hydration energy, leading to a more stable hydration layer. When confined within small pores, this stable hydration shell creates a significant steric hindrance, considerably impeding Mg^{2+} diffusion and resulting in minimal permeability. However, as pore size increases, the steric hindrance diminishes, allowing Mg^{2+} to experience a substantial increase in diffusion rate. In conclusion, the smaller size and dynamic hydration behavior of Li^+ render its diffusion across membranes relatively insensitive to pore size variations. Conversely, the larger size and stable hydration shell of Mg^{2+} make its diffusion highly sensitive to pore size changes.

Comment 6: The coupling effect between the trans-membrane ion transport of Li^+ and Cl^- via a spontaneously-arising electric field is confusing. The PA membrane is negatively charged, which shows a stronger attraction towards Mg^{2+} than Li^+ , and a repulsion towards Cl^- . Because Mg^{2+} is a dominant ion in the feed, the increasing mean pore size from 5.2 to 6.4 Å should have a remarkable effect on the Mg^{2+} permeability.

Response: Spontaneously arising electric fields in conjunction with the solution-diffusion-electromigration (SDEM) model offer a comprehensive framework for understanding ion transport across membranes in multi-ion systems (see *J. Membr. Sci.* 2013, 447, 463-476; *J. Membr. Sci.* 2017, 523, 361-372). In this study, we employed a $\text{MgCl}_2/\text{LiCl}$ salt mixture to evaluate the $\text{Li}^+/\text{Mg}^{2+}$ separation selectivity of OSARIP-derived PA membranes. This system comprises three distinct ions. In the mixture of 20:1 mass ratio of Mg^{2+} to Li^+ , the Cl^- concentration is predominantly determined by Mg^{2+} , being roughly twice its concentration. The high Cl^- concentration, coupled with its smaller size, facilitates faster transport, leading to charge imbalance across the membrane and the establishment of spontaneous electric fields. These fields, in turn, accelerate Li^+ transport, resulting in significantly lower rejection compared to single-salt (LiCl) feed, potentially even causing negative

rejection.

While increasing the PA membrane pore size inevitably reduces Mg^{2+} rejection, it also mitigates the concentration polarization effect arising from MgCl_2 accumulation. Consequently, the surface Cl^- concentration decreases, subsequently slowing its transport and weakening both the charge imbalance and the strength of the spontaneously-arising electric fields across the membrane. This weakened field strength diminishes the accelerating effect on Li^+ , leading to a less pronounced negative retention at larger pore sizes.

Response to Reviewer 3

General Comment: In this manuscript, Peng et al. presents the use of an oil-soluble surfactant, dodecyl phosphate (DDP), during interfacial polymerization (IP) to enhance the pore size uniformity, and therefore selectivity, of nanofiltration membranes for $\text{Li}^+/\text{Mg}^{2+}$ separation. Membranes for $\text{Li}^+/\text{Mg}^{2+}$ separation have gained interest due to their ability to extract lithium from geothermal brines but are currently limited by low selectivities. Previous works have found that the addition of surfactants in the aqueous phase promotes the diffusion of amine monomers across the aqueous/organic interface during IP in a process known as surfactant-assembly regulated IP (SARIP), resulting in polyamide layers with more uniform pore size distributions. In this study, Peng et al. added the surfactant DDP in the organic, rather than aqueous, phase in order to eliminate surfactant aggregation in the aqueous phase and further enhance amine diffusion in a modified organic-SARIP (OSARIP) process. They report that OSARIP resulted in a narrower pore size distribution and smaller mean pore size than IP and SARIP, resulting in over 99% rejections of divalent salts by size-sieving. Notably, in mixed-salt solutions of MgCl_2 and LiCl , OSARIP membranes demonstrated $\text{Li}^+/\text{Mg}^{2+}$ selectivities as high as 4147 due to the combined effects of high Mg^{2+} rejection from size-sieving and enhanced Li^+ transport from counter-ion coupling with Cl^- . Overall, this work presents a significant improvement in $\text{Li}^+/\text{Mg}^{2+}$ separation by implementing a simple modification to conventional IP. However, there are several details that are somewhat oversold and overinterpreted and should be addressed to make a good scientific contribution. I also believe that the authors need to be able to show what would happen under more realistic scenarios with these membranes.

Response: We appreciated the positive comment from the reviewer. We have tried our best to respond the reviewer's concern, especially for the selectivity of the PA NF membrane worked under more realistic scenarios. In practical industrial applications of NF technology for lithium extraction from brine, feed solutions typically exhibit higher concentrations and contain additional metal ions, such as Na⁺, K⁺, and Ca²⁺. To simulate the feed solution used in practical application, we used a mixed salt solution with total salt concentration of up to 10000 ppm, in which other metal ions, such as Na⁺, K⁺, and Ca²⁺, are also contained. Even under this condition, the PA NF membrane demonstrates exceptional Li⁺/Mg²⁺ selectivity more than 200 (see Table R1). This excellent performance underscores the superiority of membranes with uniform pore sizes for lithium extraction from salt lake brine.

Table R1. The concentration of Li⁺ and Mg²⁺ in the feed and permeate, respectively. The total salt concentration of the feed is up to 10000 ppm. The applied pressure is 6 bar.

Ion	Concentration (ppm)		Rejection (%)	Li ⁺ /Mg ²⁺ selectivity
	feed	permeate		
Mg ²⁺	2271	23	98.99	
Li ⁺	107	229.5	-114.49	
Na ⁺	78.2	162.5	-107.67	212
K ⁺	108	218.5	-102.31	
Ca ²⁺	21.2	undetected		

Comment 1: Absolute pore size distribution in terms of angstroms from a simple correlation of polymers that pass through a NF membrane is an oversimplification that I don't think should be highlighted so prominently in this paper. As the authors themselves mention "no steric and hydrodynamic interactions between these neutral solutes and the pores of the membranes", we know that this is not true at this scale of the pore and sugars going through the pores. Also normal hydrodynamics does not even apply. I know it would make the paper less easy to write but I really don't agree with all the size-based arguments here because of all this uncertainty. The authors talk about narrowing pore size distribution from 3.0 - 8.8 Å to 3.2 - 8.0 Å, I am not sure

that we can be confident about this at all given the methods used. However, I do agree that the pore size is definitely narrowed as seen from the sharper MW cutoff from the sugar filtration experiments.

Response: We would like to emphasize that determining the precise pore size distribution (PSD) of nanofiltration (NF) membranes remains a significant challenge. While techniques like transmission electron microscopy (TEM) and atomic force microscopy (AFM) offer valuable insights, their resolution limitations and requirement for water-saturated conditions hinder their accuracy for sub-nanometer pores. Positron annihilation lifetime spectroscopy (PALS) provides information on free volume and size distribution within the polyamide active layer, but its accessibility is restricted. Therefore, utilizing a series of charge-neutral probe molecules with varying molecular weights has emerged as a widely adopted and reliable alternative, as demonstrated by its agreement with PALS measurements (*Nat. Commun.* 2020, 11, 2015). In this article, we focus on highlighting the PSD in angstroms primarily to showcase the advantages of our PA membrane preparation method. It is important to note that while all PSDs presented here are derived from charge-neutral molecule rejection data, the comparisons between NF membranes are based on a consistent benchmark, ensuring the validity and robustness of our conclusions

Comment 2: The 4000 selectivity is a little oversold. That value only occurs at very low lithium concentration and very low total salt concentrations (2000 ppm) which is totally impractical for any brine-based lithium separations.

Response: Firstly, we wish to clarify that the use of 2000 ppm salt solution for evaluating membrane separation performance is a well-established and extensively adopted method within the field of NF research (see *Nat. Commun.* 2023, 14, 5483; *J. Membr. Sci.* 2023, 647, 121515; *J. Membr. Sci.* 2022, 663, 121027; *J. Membr. Sci.* 2021, 635, 119054, and so on). This method offers a primary assessment of the ion separation performance of NF membranes and allows for the comparison of different membranes under standardized conditions. Moreover, the use of this benchmark makes comparing the performance of NF membranes convenient, facilitating collaboration and progress in the field.

Secondly, despite the acknowledged advantage of a higher mass ratio of Mg^{2+}/Li^{+} for achieving enhanced selectivity, our membrane demonstrates an

exceptional selectivity of up to 800 for the separation of Li^+ and Mg^{2+} , even when the mass ratio of $\text{Mg}^{2+}/\text{Li}^+$ is set at 20:1. This achievement markedly surpasses numerous values reported in the current literatures. Upon elevating the total salt concentration to 5000 ppm, our membrane consistently maintains a separation selectivity of 185 for Li and Mg. This result is still excellent, representing the highest value among the reported values. In order to provide a more comprehensive illustration of the separation selectivity, we conducted supplementary experiments with a further increase in salt concentration to 10000 ppm. At this concentration, our findings reveal a Li^+ retention of -102.8% and Mg^{2+} retention of 99.27%, resulting in a $\text{Li}^+/\text{Mg}^{2+}$ separation selectivity of 277, underscoring the remarkable efficacy of our membrane in these experimental conditions.

Comment 3: Further there is a dramatic impact of salinity on lithium selectivity, I then really question the authors' assertion that the selectivity is solely size based. Their further assertion of chloride coupled high transport of lithium is also suspect in the light of a mixed salt with NaCl as described below.

Response: As pointed out by the reviewer, the salinity significantly impacts the Li/Mg selectivity of PA membrane, with pore size not being the sole factor affecting selectivity. However, we would like to emphasize that a uniform pore structure is a fundamental prerequisite and essential guarantee for achieving high-selectivity separation. The PA membrane prepared through OSARIP possesses extremely small and uniform pores, providing a sufficiently high rejection for Mg^{2+} , which is a fundamental condition for achieving high selectivity. On this basis, the mass ratio of $\text{Mg}^{2+}/\text{Li}^+$ and the total salt concentration in the mixed salt solution will lead to variations in Li^+ retention, thereby influencing selectivity. We conducted detailed studies and discussions on this point in the manuscript, with experimental results presented in Figures 3a and 3b. In details, high $\text{Mg}^{2+}/\text{Li}^+$ ratios and elevated salt concentrations both induce more negative Li^+ retention. This negative rejection is primarily attributed to the transport coupling of cation and anion with low rejection during the separation process. This phenomenon is common in the separation of multi-ion systems and has been thoroughly explained using the SDEM model by the membrane community (see *J. Membr. Sci.* 2001, 185, 223; *Chem. Eng. Sci.* 2013, 104, 1107; *Desalination* 2019, 468, 114081; *J. Membr. Sci.* 2021, 620, 118862; *Nat. Water* 2023, 1, 291). Therefore, if the solution contains NaCl, coupling occurs between Na^+

and Cl^- as well, resulting in negative rejection for Na^+ . To demonstrate this, we used a mixed salt solution with a total salt concentration of more than 10000 ppm as the feed, containing MgCl_2 , LiCl , NaCl , KCl , and CaCl_2 , as presented in Table R1. Experimental results showed negative retention for all monovalent cations, namely Li^+ (-114.49%), Na^+ (-107.67%), and K^+ (-102.31%).

Table R1. The concentration of Li^+ and Mg^{2+} in the feed and permeate, respectively. The total salt concentration of the feed is up to 10000 ppm. The applied pressure is 6 bar.

Ion	Concentration (ppm)		Rejection (%)	$\text{Li}^+/\text{Mg}^{2+}$ selectivity
	feed	permeate		
Mg^{2+}	2271	23	98.99	
Li^+	107	229.5	-114.49	
Na^+	78.2	162.5	-107.67	212
K^+	108	218.5	-102.31	
Ca^{2+}	21.2	undetected		

Comment 4: I also wanted to comment on this paragraph from the perspective of what is seen and what is actually applicable in a realistic scenario.

Lines 270 -274 “Specifically, MgCl_2 is the dominant salt in the simulated brine and the concentration of Cl^- is mainly determined by the MgCl_2 concentration. Intercepting Mg^{2+} ions can lead to the accumulation of Cl^- ions near the membrane surface. As Mg^{2+} rejection increases, the concentration gradient of Cl^- across the membrane increases, resulting in the faster transport of Cl^- . Accompanying the transport of Cl^- , the transport of the counter-ion with equivalent charge is necessary to maintain charge neutrality in the permeate, thus establishing the spontaneous potential gradient. Considering the larger steric hindrance of Mg^{2+} for partition into the pores of the membrane, accelerating the Li^+ transport is the only choice.” Except Lithium is definitely not the only choice, in an actual brine there will be much more sodium which has an even hydration energy and about the same size of lithium. Thus exaggerated lithium selectivity seen here will not occur in any realistic scenario. This should be acknowledged prominently. Further, there should be at least one set of

experiments with actual brine type composition of the feed stream to see how this membrane would perform under more realistic scenarios where these anomalies with single and two salts mixed together are diluted away.

Response: Firstly, we would like to clarify that the PA membrane prepared from OSARIP is specifically employed for the selective separation of Li^+ and Mg^{2+} . The term "selectivity" in this context pertains to the capability of the membrane to separate Li^+ and Mg^{2+} , rather than differentiating Li^+ from other metal ions such as Na^+ and K^+ . In the extraction of lithium from salt lake brine, one of the key process is the separation of Li^+ and Mg^{2+} ions, as the ions with higher abundance including Na^+ and K^+ are precluded through solar-driven evaporation-induced precipitation (see refs. *Environ. Sci.: Water Res. Technol.* 2017, 3, 593-597; *J. Membr. Sci.* 2021, 635, 119441; *Desalination* 2019, 468, 114081; *Nat. Water* 2023, 1, 291.). Although trace amounts of Na^+ , K^+ , and Ca^{2+} may persist in the brine, the final isolation of Li^+ could be achieved through precipitation with Na_2CO_3 , yielding LiCO_3 . Therefore, Na^+ in the permeate has little effect on the precipitation process.

In this study, a binary mixture comprising MgCl_2 and LiCl was employed to simulate brine, allowing us to investigate the selective separation performance of the resulting PA membrane in separating Li^+ and Mg^{2+} . The exceptional Li/Mg selectivity presented by the PA membrane is attributed to its extremely small pore size and uniform pore size distribution, facilitating an ultrahigh rejection of Mg^{2+} . It is real and not overstated.

Addressing the impact of Na^+ on selectivity, we did a supplementary experiment by using a simulated brine containing 2271 ppm Mg^{2+} , 107 ppm Li^+ , 78.2 ppm Na^+ , 108 ppm K^+ , and 21.2 ppm Ca^{2+} as the feed (with a total salt concentration of approximately 10000 ppm) (Table R1). ICP measurements of ion concentrations revealed the rejection of 98.99% for Mg^{2+} , -114.49% for Li^+ , and -107.67% for Na^+ . The corresponding Li/Mg selectivity was measured at 212. This result demonstrates that the transport of Cl^- can enhance the transport of Na^+ . Furthermore, the presence of Na^+ was found to exert minimal influence on the selectivity of the PA membrane prepared from OSARIP in the selective separation of Li^+ and Mg^{2+} .

Table R1. The concentration of Li^+ and Mg^{2+} in the feed and permeate, respectively. The total salt concentration of the feed is up to 10000 ppm. The applied pressure is 6 bar.

Ion	Concentration (ppm)		Rejection (%)	Li ⁺ /Mg ²⁺ selectivity
	feed	permeate		
Mg ²⁺	2271	23	98.99	
Li ⁺	107	229.5	-114.49	
Na ⁺	78.2	162.5	-107.67	212
K ⁺	108	218.5	-102.31	
Ca ²⁺	21.2	undetected		

Comment 5: The high negative rejection for lithium is less believable for me. Did the authors account for concentration polarization at all. It seems to me that the negative rejection goes up as the concentration of the Lithium goes down. Essentially there is less concentration polarization and thus less passage. Assuming a concentration polarization of 1.5-2x (which is not unreasonable based on crossflow rates), from table S4, none of the rejection values would be negative if the membrane concentration is used instead of the feed concentrations.

Response: First, the rejection is directly calculated according to the ion concentration in the feed and the permeate. This way of calculating retention is more in line with what happens when membranes are actually used. Second, if the reviewer is familiar with the application of PA NF membranes in the separation of multi-ion solution systems, they will find that the negative rejection phenomenon is common and have been well-explained by the researchers through the transport theory of solution-diffusion-electro-migration (SDEM) model (see refs. *J. Membr. Sci.* 2017, 523, 361-372; *J. Membr. Sci.* 2013, 447, 463-476). Similar phenomenon has been also observed in many publications of using NF for extracting lithium from salt lake brine recently (see refs. *J. Membr. Sci.* 2022, 663, 121027; *Desalination* 2019, 468, 114081; *Nat. Water* 2023, 1, 291; *J. Membr. Sci.* 2024, 690, 122207;). Negative rejection only occurs in the separation of mixed salt solutions due to the coupling interaction between cations and anions with low rejection in the solution. For example, the PA membrane we prepared has a high rejection for Mg²⁺. However, Cl⁻ and Li⁺ are difficult to retain by the membranes due to their smaller ionic size. In the mixed salt solution, the Cl⁻ concentration primarily depends on the concentration of MgCl₂ due to large mass ratio of Mg²⁺/Li⁺ (20-60), which is much higher than that of Li⁺. The

higher concentration of Cl^- facilitates its faster trans-membrane transport during the separation process. The Cl^- transport further pulls Li^+ along to maintain charge neutrality, accelerating Li^+ transport. Consequently, increasing the $\text{Mg}^{2+}/\text{Li}^+$ ratio in the solution also increases Cl^- concentration, further enhancing its transport and leading to more pronounced negative Li^+ rejection. Similarly, increasing the total mixed salt concentration promotes negative Li^+ rejection.

Comment 6: To confirm reproducibility, I would like to ask authors to include information on the number of membrane replicates made, number of trials, and error bars both in the main text and the supplementary information.

Response: We have provided the number of membrane replicates in the main text (page 18, line 13 and page 19, line 10) and revised supplementary information. Each measurement was conducted with at least three membrane replicates.

Comment 7: The authors reference the supplementary information for additional details and sources in the figures that could be included in the main text as well. Specifically, I would like to ask the authors cite the references for the data points shown in Figures 2-3. Additionally, Figure 4a would benefit from added details differentiating the eight prepared membranes, especially as M4 and M8 both are described as “DDP 0.1 g/L” yet exhibit significantly different pore size ranges.

Response: The data points used for comparison in Figure 2-3 are sourced from 69 references, readily accessible alongside their corresponding references in the Supplementary Tables 2, 7, 8 in the Supplementary information. Citing all these references within the main text would exceed the 70-reference limit set by *Nature Communications*.

For the preparation of M4 and M8 membranes, their PIP concentrations are 2.5 g/L and 5 g/L, respectively. To address the reviewer's suggestion and avoid any potential misunderstandings, more detailed preparation information for all eight membranes have been added into the Figure 4a in the revised manuscript. The revised Figure 4 is shown below.

Revised Figure 4

Comment 8: On page 5, line 145, authors mention that the OSARIP membrane displayed a smoother surface compared to the IP one, referencing the surface and cross-sectional images shown in Figure 1. I would like to ask the authors if they have any further characterization of membrane roughness to justify this claim.

Response: Upon comparing surface SEM images of PA NF membranes prepared with OSARIP and traditional IP, we observed the smaller convex features on the surface of the OSARIP resulted membrane. In response to the reviewer's suggestion, we further investigated surface roughness using AFM. As shown in Figure R8, the AFM result confirms the presence of smaller convex structures on the surface of OSARIP resulted membrane. However, the Ra and Rq roughness values, which quantify surface roughness, show negligible differences between the two membranes. Therefore, based on the AFM results, it is appropriate to remove the description of a "smoother surface" from the revised manuscript to avoid potential misinterpretations.

Comment 9: On pages 10-11, lines 289-300, the authors compare negatively-charged OSARIP membranes made by piperazine (PIP) to positively-charged ones made by polyethylene imine (PEI) to justify the claim that a negatively-charged surface enhances the $\text{Li}^+/\text{Mg}^{2+}$ selectivity of OSARIP membranes. However, membranes made by PEI tend to have looser structures compared to those made by PIP, and this

was evident in the pore size distribution curves and performance data presented in the SI that show that PEI OSARIP membranes had slightly higher pore sizes, wider pore size distributions, and lower Mg^{2+} rejections than those made by PIP [1]. The authors suggested that the higher rejection of Li^+ by PEI membranes was caused by increased Donnan exclusion from the positively-charged surface, but it seems possible that the higher Li^+ rejection could've also been caused by a decreased coupling effect from lower Mg^{2+} rejections. It is thus not clear whether the difference in performance between PIP and PEI-based membranes should be attributed to their surface charges or difference in structures. Given this, would the authors be able to provide further experimental or simulation evidence to validate the claim that negatively-charged OSARIP membranes demonstrate higher selectivities?

Response: The PA membrane prepared using PEI monomers is used as comparison to prove the impact of Cl^- transport on the rejection of Li^+ . Although the PA membranes prepared using PEI as monomer have looser structures, we prepared five PA membranes with varying pore sizes to better analyze the impact of chemical structure on ion transport. Using the same mixed salt solution as feed, we found that PEI-based membranes exhibited significantly higher Li^+ rejection than PIP-based ones. And the increase in Mg^{2+} rejection paralleled the rise in Li^+ rejection, unlike the PIP-based membranes. This strongly suggests that the higher Li^+ rejection in PEI membranes is not solely due to a weakened coupling effect from lower Mg^{2+} rejection.

To explain this enhanced Li^+ rejection, we propose that the chemical environment within the membrane pores interacts with ions, influencing their transport. Compared to PIP-based membranes, PEI-based membranes possess a higher density of amine groups, leading to a strong positive charge. This abundance of amine groups strongly interacts with Cl^- ions, hindering their transport and contributing to the Donnan exclusion of Li^+ . As a result, PEI-based membranes display higher Li^+ rejection than PIP-based membranes. It is important to note that PEI-based membranes are primarily used here as a reference to highlight the superiority of negatively charged PA membranes with narrower pore size distributions for Li^+ extraction. While the transmembrane transport of ions is complex and requires further investigation through detailed experiments and simulations which will be explored in future studies, this work focuses on the impact of uniform pore size distribution on Li/Mg selectivity.

Comment 10: One of the main findings in this work was that a concentration of 0.1 g/L DDP corresponded to the lowest interfacial tension as well as smallest mean pore size and distribution relative to both lower and higher concentrations. Interestingly, membranes with 0.1 g/L DDP also demonstrated both higher rejections and permeances relative to other concentrations, as shown in Figure S5, but this is not discussed in detail in the manuscript. I would like the authors to comment on the effect of surfactant concentration on OSARIP membranes.

Response: Through the electrostatic interaction between the phosphonic group in DDP and the amine group in PIP, the DDP monolayer can adsorb the PIP molecules to increase their concentration near the interface, which is benefit for accelerating the trans-interface diffusion of PIP by enlarging the PIP concentration difference between the water and hexane. When the concentration of DDP in hexane is 0.1 g/L, the interfacial tension of hexane/water was the lowest, indicating the formation of the densest molecular layer. In this case, the concentration of PIP near the interface is the highest. Further increasing the concentration of DDP will destroy the molecular structure of the assembly at the oil/water interface, causing the decrease of PIP concentration near the interface. According to the reviewer's suggestion, we have added the statement of ".....PIP concentration near the oil/hexane interface, reaching its highest point at a DDP concentration of 0.1 g L⁻¹" in the revised manuscript (page 5 line 3-5).

Reviewers' Comments:

Reviewer #1:

Remarks to the Author:

The authors have made efforts and revised the manuscript. They further performed additional experiments. Generally, they have addressed most questions. The revised manuscript is suitable for publication in Nature Communications.

After reading the other reviewers' comments, I have some more minor suggestions that might help the authors improve the quality and readability of their manuscript.

1. It is indeed confusing to use rejection to evaluate the selectivity, especially given that the monovalent ion rejection becomes negative in the mixtures. The authors can also consider calculating the permeation rates of Li and Mg based on the water flux and salt concentration, so they can derive the ion permeation rates. Comparing the ion permeation rates might make it easier for the readers to understand the results (especially for those outside the membrane field).

2. I am still curious how the concentration changed as a function of time for both permeate and feed solutions. As explained by the authors, the concentration did change significantly with the water recovery ratio (Figure R4, R5). However, the data presented is a bit confusing. Perhaps the authors can plot the change of concentration as a function of time too.

3. Could the authors justify the composition of the simulated brine solution (Table 6)? It would be better to use the composition of a typical salt lake. The concentrations of Na and K also seem to be low. Could the authors perform additional experiments with relatively high concentrations of Na and K?

2. I agree with Reviewer 3 that the authors should be cautious with the record-high selectivity. To some extent, it is a bit overselling to emphasize the selectivity of 4000 obtained from binary mixtures in the abstract. In reality, the selectivity obtained from real brine mixtures is more important. As demonstrated by the authors, the Li/Mg selectivity decreased evidently when tested with simulated brine solution.

Reviewer #2:

Remarks to the Author:

In the revised manuscript, the authors provided detailed explanations to address my concerns. The revised version is recommended for publication.

Reviewer #3:

Remarks to the Author:

The revisions to this manuscript have improved clarity and validate the findings of this work. The follow-up experiment using simulated brine conditions, added experimental details, and adjusted conclusions based on additional characterization are appreciated. I recognize that direct measurement of pore size may be outside the scope of this work, and using the neutral solute rejection method to determine the pore size distribution of OSARIP, SARIP, and commercial membranes allows for direct comparison. Similarly, I accept that while reporting selectivity based on measured, local feed and permeate concentrations at lower salinity may not be representative of a realistic module-scale process, it is standard in the literature.

However, the results of the simulated brine experiment are somewhat unexpected, as the authors reported a selectivity of 212, higher than that reported for the original 3000-5000 ppm experiments consisting solely of LiCl and MgCl₂. I would expect the presence of interfering monovalent salts to decrease Li⁺ passage and the higher salinity to reduce Mg²⁺ rejection (as seen in the increasing salinity experiments up to 5000 ppm), yet this is not the case here – would the authors be able to comment on these results?

Though metrics such as concentration polarization-adjusted rejection and lithium recovery and

additional analysis on the simulated brine experiment would demonstrate the practical applicability of the membranes presented in this work, we believe that the results represent an improvement from membranes reported in the literature. Overall, the OSARIP membranes presented in this paper demonstrate significantly improved Li/Mg selectivity relative to existing nanofiltration membranes, and the revisions help to validate these results.

Response to Reviewer 1

The authors have made efforts and revised the manuscript. They further performed additional experiments. Generally, they have addressed most questions. The revised manuscript is suitable for publication in Nature Communications. After reading the other reviewers' comments, I have some more minor suggestions that might help the authors improve the quality and readability of their manuscript.

1. It is indeed confusing to use rejection to evaluate the selectivity, especially given that the monovalent ion rejection becomes negative in the mixtures. The authors can also consider calculating the permeation rates of Li and Mg based on the water flux and salt concentration, so they can derive the ion permeation rates. Comparing the ion permeation rates might make it easier for the readers to understand the results (especially for those outside the membrane field).

Response: We appreciate the reviewer's suggestion. Regardless of the methodology employed to calculate selectivity, whether through the rejection of Li⁺ and Mg²⁺, or their permeation rates, or the concentrations of these ions in both the feed and permeate streams, the resulting outcome remains consistent. In the revised supporting information (see equation S3), we provide the calculation equation for Li⁺/Mg²⁺ selectivity based on all three methods, along with a detailed derivation process that elucidates why these approaches yield identical result. The revised equation and derivation are presented below.

$$S_{Li,Mg} = \frac{\frac{J_{Li}}{C_{f,Li}}}{\frac{J_{Mg}}{C_{f,Mg}}} = \frac{\frac{J_w \cdot C_{P,Li}}{C_{f,Li}}}{\frac{J_w \cdot C_{P,Mg}}{C_{f,Mg}}} = \frac{\frac{C_{P,Li}}{C_{f,Li}}}{\frac{C_{P,Mg}}{C_{f,Mg}}} = \frac{1 - \left(1 - \frac{C_{P,Li}}{C_{f,Li}}\right)}{1 - \left(1 - \frac{C_{P,Mg}}{C_{f,Mg}}\right)} = \frac{1 - R_{Li}}{1 - R_{Mg}} \quad (S3)$$

2. I am still curious how the concentration changed as a function of time for both permeate and feed solutions. As explained by the authors, the concentration did change significantly with the water recovery ratio (Figure R4, R5). However, the data presented is a bit confusing. Perhaps the authors can plot the change of concentration as a function of time too.

Response: We would like to elucidate the rationale behind relating the concentration

changes of Li^+ and Mg^{2+} in the feed and permeate solutions to the water recovery ratio, rather than filtration time. The concentration changes of Li^+ and Mg^{2+} in the feed and permeate solutions are primarily driven by the volume reduction of the feed solution during filtration, rather than directly by the filtration time itself. As Li^+ exhibits a negative rejection, signifying its faster migration compared to water, as the permeate volume increases, translating to a higher water recovery ratio, the loss of Li^+ from the feed solution intensifies, resulting in a decline in its concentration. This phenomenon is clearly illustrated in Fig. 4Ra, where the Li^+ concentration in the feed solution exhibits a significant decrease with increasing water recovery rate.

Furthermore, industrial applications commonly utilize membranes in modular configurations. In these settings, the water recovery ratio emerges as a critical parameter governing filtration performance. Compared to filtration time, exploring the relationship between the solute composition of the feed and permeate solutions and the water recovery ratio holds greater practical relevance. Consequently, our investigation prioritized elucidating the impact of varying water recovery ratios on the concentration profiles of Li^+ and Mg^{2+} , while omitting the collection of filtration time data.

Figure 4Ra. The concentration of Mg^{2+} and Li^+ in the feed as a function of water recovery ratio.

3. Could the authors justify the composition of the simulated brine solution (Table 6)? It would be better to use the composition of a typical salt lake. The concentrations of Na and K also seem to be low. Could the authors perform additional experiments with relatively high concentrations of Na and K?

Response: While the simulated brine composition presented in Table 6 may not perfectly replicate real brines, it nonetheless incorporates the primary ionic constituents found in real brines. This simulated brine serves as a suitable feed solution for evaluating the performance stability of the polyamide nanofiltration (PA NF) membrane fabricated using the OSARIP method for selectively separating Li^+ and Mg^{2+} . This is because the selective separation of Mg^{2+} and Li^+ constitutes a critical step in lithium extraction from brine, as other more abundant ions (such as Na^+ , K^+ , and Ca^{2+}) are pre-removed through solar-driven evaporation. Therefore, the concentration of Na^+ and K^+ in lithium extraction brines are usually close to that of Li^+ . In fact, the total salt concentration is the most significant factor influencing the selective performance of the PA NF membrane. To investigate this effect, we investigated the simulated brine with the total salt concentration as high as 10,000 ppm, which involves Na^+ and K^+ with the concentrations comparable to Li^+ . Our experimental results demonstrate that the PA NF membrane prepared via OSARIP exhibits stable performance in differentiating Li^+ and Mg^{2+} , even under high salinity conditions. The co-presence of Na^+ and K^+ has a negligible impact on selectivity. This can be attributed to the inherent minimal selectivity of PA NF membranes towards monovalent cations like Na^+ , K^+ , and Li^+ . During the separation process, the transport behavior of Na^+ and K^+ closely mirrors that of Li^+ . This allows us to essentially consider Na^+ and K^+ as equivalent solutes to Li^+ . Consequently, elevated concentrations of Na^+ and K^+ in the feed do not significantly alter the selective separation performance of Li^+ and Mg^{2+} by the PA NF membrane prepared from OSARIP.

4. I agree with Reviewer 3 that the authors should be cautious with the record-high selectivity. To some extent, it is a bit overselling to emphasize the selectivity of 4000 obtained from binary mixtures in the abstract. In reality, the selectivity obtained from real brine mixtures is more important. As demonstrated by the authors, the Li/Mg selectivity decreased evidently when tested with simulated brine solution.

Response: According to the reviewer's comment, we have changed the description of "Under the solely size sieving effect, an unprecedentedly high Mg^{2+} rejection rate of

99.96% and $\text{Li}^+/\text{Mg}^{2+}$ selectivity over 4000 are achieved. This value is one to two orders of magnitude higher than all the currently reported pressure-driven membranes, and even higher than the microporous framework materials, including COFs, MOFs, and POPs.” to “Under the solely size sieving effect, an exceptional Mg^{2+} rejection rate of over 99.9% is achieved. This results in an exceptionally high $\text{Li}^+/\text{Mg}^{2+}$ selectivity, which is one to two orders of magnitude higher than all the currently reported pressure-driven membranes, and even higher than the microporous framework materials, including COFs, MOFs, and POPs.” in the revised manuscript (page 2, line 1-5).

Response to Reviewer 3

The revisions to this manuscript have improved clarity and validate the findings of this work. The follow-up experiment using simulated brine conditions, added experimental details, and adjusted conclusions based on additional characterization are appreciated. I recognize that direct measurement of pore size may be outside the scope of this work, and using the neutral solute rejection method to determine the pore size distribution of OSARIP, SARIP, and commercial membranes allows for direct comparison. Similarly, I accept that while reporting selectivity based on measured, local feed and permeate concentrations at lower salinity may not be representative of a realistic module-scale process, it is standard in the literature.

However, the results of the simulated brine experiment are somewhat unexpected, as the authors reported a selectivity of 212, higher than that reported for the original 3000-5000 ppm experiments consisting solely of LiCl and MgCl_2 . I would expect the presence of interfering monovalent salts to decrease Li^+ passage and the higher salinity to reduce Mg^{2+} rejection (as seen in the increasing salinity experiments up to 5000 ppm), yet this is not the case here – would the authors be able to comment on these results?

Though metrics such as concentration polarization-adjusted rejection and lithium recovery and additional analysis on the simulated brine experiment would demonstrate the practical applicability of the membranes presented in this work, we believe that the results represent an improvement from membranes reported in the literature. Overall,

the OSARIP membranes presented in this paper demonstrate significantly improved Li/Mg selectivity relative to existing nanofiltration membranes, and the revisions help to validate these results.

Response: We are grateful to the reviewer for accepting our revision to the manuscript. The reviewer raised a pertinent query regarding the seemingly anomalous separation selectivity of 212 achieved by the OSARIP resulted PA NF membrane using a simulated brine with 10000 ppm total salt concentration as feed, particularly in the presence of Na^+ and K^+ . We would like to clarify that the PA NF membranes prepared from OSARIP exhibit a highly uniform pore size. This enables the membrane to show high rejection to Mg^{2+} based on size-sieving mechanism. As size-sieving is primarily a physical phenomenon governed by steric hindrance, it is very slightly affected by external factors such as the total salt concentration. Therefore, the membrane retains its high rejection of Mg^{2+} even when the total salt concentration is as high as 10,000 ppm. In addition, increasing the ion concentration is beneficial to the transport of Li^+ through the membrane due to increased concentration polarization force. That is to say, the higher the ion concentration, the more leads to a negative rejection of Li^+ . While the transport behaviors of Na^+ and K^+ ions in the feed solution is similar to that of Li^+ ions. The presence of Na^+ and K^+ ions has almost no influence on the transport of Li^+ through the membrane. Due to the above reasons, the OSARIP resulted PA NF membrane using a simulated brine with 10000 ppm total salt concentration as feed in the presence of Na^+ and K^+ shows higher selectivity than that reported for the original 3000-5000 ppm experiments consisting solely of LiCl and MgCl_2 .

Reviewers' Comments:

Reviewer #1:

Remarks to the Author:

The authors have satisfactorily addressed the remaining concerns. The manuscript is suitable for publication.

Reviewer #3:

Remarks to the Author:

The authors have done a good job of responding to my comments. I have no further comments. This is an excellent contribution to the literature in this important area.